# Reference-Free Meta-Learning for Generalized Implicit Neural Representation in Efficient MRI Reconstruction

**Haonan Zhang** [1]  **Qing Wu** [2]  **Xuanyu Tian** [2]  **Bowen Li** [1]  **Yuyao Zhang** [2]  **Hongjiang Wei** [1 3]

## Abstract

Implicit Neural Representation (INR) has emerged as a powerful paradigm for continuous MRI reconstruction. However, standard self-supervised INR requires time-consuming optimization from scratch for each scan, hindering clinical deployment. This work presents IPOD, a Reference-Free Meta-Learning framework designed to learn generalized parameter initializations for INR directly from undersampled data. Distinct from conventional meta-learning that relies on fully-sampled ground truth, IPOD operates in an inverse-problem-driven manner, leveraging diverse reconstruction tasks with varying sampling patterns to capture a robust prior. Furthermore, we introduce an adaptive meta-update strategy modulated by task-specific performance to ensure optimal parameter distribution for diverse anatomical structures. Extensive experiments demonstrate that IPOD provides a superior initialization that enables rapid adaptation and achieves high-fidelity reconstruction across various imaging protocols, significantly outperforming existing INR baselines. By eliminating the dependence on reference images, IPOD offers a scalable and efficient solution for a wide range of imaging inverse problems. Code and data available at: https://github.com/zhn00310/RFML4MRI

---

[1]School of Biomedical Engineering, Shanghai Jiao Tong University, Shanghai, China [2]School of Information Science and Technology, ShanghaiTech University, Shanghai, China [3]National Engineering Research Center of Advanced Magnetic Resonance Technologies for Diagnosis and Therapy (NERC-AMRT), Shanghai Jiao Tong University, Shanghai, China. Correspondence to: Hongjiang Wei <hongjiang.wei@sjtu.edu.cn>.

*Proceedings of the 43rd International Conference on Machine Learning*, Seoul, South Korea. PMLR 306, 2026. Copyright 2026 by the author(s).

## 1. Introduction

Magnetic resonance imaging (MRI) offers superior soft tissue contrast, making it essential for clinical diagnosis and neuroscience research (Feng et al., 2023). However, its long acquisition times lead to higher healthcare costs and patient discomfort. Accelerating MRI acquisition is commonly achieved by collecting undersampled $k$-space data, but reconstructing high-quality images under such conditions is challenging due to violations of the Nyquist sampling theorem. With the advancement of deep learning (DL), supervised reconstruction methods have shown remarkable improvements by learning mappings from corrupted to artifact-free MR images (Sun et al., 2020; Han et al., 2018; Aggarwal et al., 2019; Qin et al., 2019; Wang et al., 2022; 2023b;a; Song et al., 2025). Nevertheless, these approaches require large-scale fully sampled data, which is impractical, and often suffer from poor generalization under domain shifts at inference.

Recently, implicit neural representations (INRs) have emerged as a self-supervised paradigm to solve inverse problems in medical imaging (Shen et al., 2022; Xu et al., 2023; Spieker et al., 2023). In accelerated MRI reconstruction, INR-based methods represent the desired MR image as a continuous function and integrate the physical forward model to achieve comparable performance to supervised methods using only undersampled $k$-space data. However, existing INR approaches face two key limitations: unstable performance at high acceleration factors, and slow reconstruction speeds—particularly in 3D reconstruction scenarios, where reconstruction may take several hours (Feng et al., 2023).

Previous works have demonstrated that efficient initialization enables faster INR convergence on unseen data, with meta-learning emerging as a fundamental framework for learning optimal initialization strategies across multiple tasks (Bauer et al., 2023; Dupont et al., 2022; Tancik et al., 2021). However, these meta-initialized INRs are predominantly applied to computer vision tasks such as image regression and novel view synthesis, which rely on the availability of high-quality natural images. While recent efforts have introduced meta-learning to the medical domain (Dannecker et al., 2025; Friedrich et al., 2025; Paolis et al., 2024),

### Single-Problem Optimization

*Figure 1.* Given a single undersampled $k$-space data $\boldsymbol{S}$, the INR networks, parametrized by $\boldsymbol{\Theta} = (\theta, \phi)$, take discrete coordinates $\boldsymbol{x} = (x, y)$ as input and produce the predicted real and imaginary values of the underlying MR image. Based on the forward physical model, the loss function $\mathcal{L}$ is minimized for optimization of network parameters $\boldsymbol{\Theta}$.

they either still necessitate high-quality, fully-sampled reference data, which is resource-intensive to acquire, or focus on post-processing tasks such as super-resolution. These approaches remain distinct from the pursuit of more demanding ill-posed reconstruction inverse problems, which involve recovering high-fidelity images directly from sparse, undersampled raw data.

In this work, we propose **IPOD**, an inverse-problem-driven meta-learning framework that learns effective initialization of INR networks for MRI reconstruction. Our key innovation lies in constructing a novel meta-learning framework to efficiently meta-initialize INR networks without the dependency on high-quality fully-sampled data. *Conceptually, leveraging the physical forward model, our approach transforms the inner loop optimization to focus on solving diverse reconstruction inverse problems rather than simple image regression tasks. This physics-informed meta-learning strategy enables the framework to capture the real-world reconstruction processes, leading to more informed and robust initialization parameters.* Furthermore, we introduce an adaptive weighting mechanism that prevents the meta-learning process from being corrupted by suboptimal solution spaces. This mechanism dynamically ensures the stability and effectiveness of meta-initialized INR networks across diverse imaging scenarios and acceleration factors.

In the experiments, we adopt the fastMRI (Knoll et al., 2020) dataset for the IPOD meta-learning training phase, and comprehensively evaluate on out-of-domain datasets, including the public MoDL (Aggarwal et al., 2018) dataset and prospectively undersampled in-house data. The results consistently demonstrate rapid convergence across diverse out-of-domain scenarios, including different subjects, contrast mechanisms, sampling patterns, and forward models. Notably, IPOD is a unified meta-learning framework that provides effective initialization for various INR methods. To validate its effectiveness, we evaluate it on three representative INR frameworks.

The main contribution of this work lies in exploring the feasibility of a reference-free meta-learning method for imaging inverse problems. Specifically, our proposed approach, IPOD, learns to extract robust priors directly from undersampled measurements, thereby eliminating the conventional dependency on fully-sampled reference data. We demonstrate that IPOD serves as a versatile and powerful initialization strategy across various INR frameworks, consistently achieving significantly faster convergence and superior reconstruction fidelity across diverse contrasts, sampling patterns, and real-world scanner environments.

## 2. Related Works

**INR for MRI Reconstruction** Implicit Neural Representation (INR) has emerged as a powerful self-supervised deep learning framework for accelerated MRI reconstruction (Feng et al., 2023; Shen et al., 2022; Xu et al., 2023; Kunz et al., 2023; Feng et al., 2022; Spieker et al., 2023; Wu et al., 2021; Huang et al., 2023; Chen et al., 2023; Catalán et al., 2023; Wu et al., 2026). It represents the desired MR image as a continuous function of spatial coordinates, approximated by a neural network. By integrating the MRI physical forward model and the network's learning bias (Rahaman et al., 2019; Xu et al., 2019), INR methods can reconstruct high-quality MR images from only undersampled $k$-space data. However, most INR approaches rely on scan-specific training, which fails to exploit valuable data-driven priors. This limitation is a primary reason for their unstable performance and relatively slow reconstruction speeds, as each new dataset requires training from scratch, and the learned representations are difficult to transfer even to similar data domains. In contrast, we propose a promising direction: integrating data-driven priors via meta-learning. This approach enables us to find a well-initialized INR, which significantly enhances both reconstruction quality and speed.

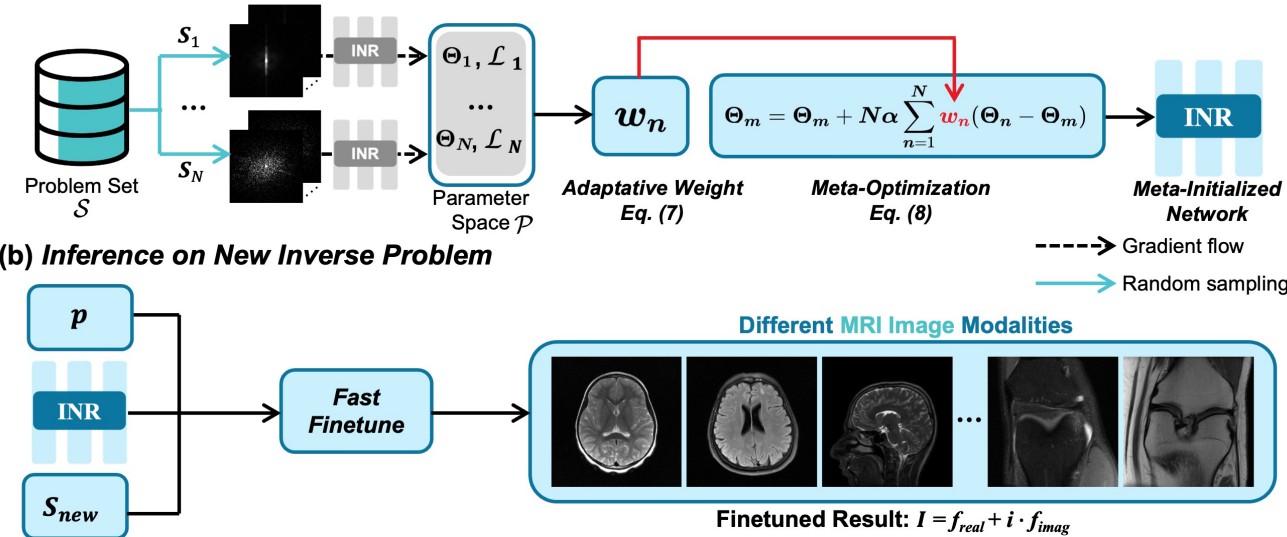

*Figure 2.* Overview of IPOD framework. In the inverse-problem-driven meta-learning procedure, a subset is first randomly sampled from the task set $\mathcal{S}_{sub} = \{\boldsymbol{S}_1, \boldsymbol{S}_2, \ldots, \boldsymbol{S}_N\} \subseteq \mathcal{S}$. After completing optimization, as shown in Fig. 1, for each single task $\boldsymbol{S}_n \in \mathcal{S}_{sub}$, adaptive weights are computed, which are used to generate efficient optimization directions in meta-optimization $\boldsymbol{\Theta}_m$. Ultimately, these meta-initialized INR networks are capable of performing fast and high-quality reconstruction across various MRI inverse problems.

**Meta-Learning** Meta-learning addresses fast adaptation and generalization with few samples, where a meta-learner is trained to quickly adapt to new tasks (Sung et al., 2018; Vinyals et al., 2016; Ravi & Larochelle, 2017; Snell et al., 2017; Mishra et al., 2018). The popular branches are optimization-based meta-learning methods like MAML (Finn et al., 2017) and Reptile (Nichol et al., 2018), which aim to find strong weight initializations that allow efficient adaptation to unseen tasks within a few optimization steps. Recent advancements have connected meta-learning with INRs, extending its possibilities to learn functions that represent data (Sitzmann et al., 2020a; Tancik et al., 2021). Moreover, the concept of using meta-learning for INR initialization has been explored in various works (Lee et al., 2021; Chen & Wang, 2022; Vyas et al., 2024; Lee et al., 2023; Kim et al., 2023; Gu et al., 2023; Yang et al., 2025). However, these methods are difficult to directly apply to inverse problems in MRI reconstruction. First, most of them require large amounts of high-quality training data, which is challenging to collect in medical imaging. Second, they typically lack integration of the physical forward model, which is essential for simulating real-world data acquisition processes and ensuring high-fidelity MRI reconstruction.

## 3. Proposed Method

### 3.1. Preliminaries

**MRI Inverse Problem** Multi-coil MRI reconstruction aims to recover an unknown MR image from under-sampled

$k$-space measurements, which is typically an ill-posed problem due to the violation of the Nyquist sampling theorem. The under-sampled $k$-space data $\boldsymbol{S}_j$ can be written as:

$$\boldsymbol{S}_j = \mathbf{M}\mathcal{F}\mathbf{C}_j\boldsymbol{I} + \boldsymbol{e}_j, \quad j \in \{1, 2, \ldots, N\}, \quad (1)$$

where $\boldsymbol{I}$ denotes the desired MR image, $\boldsymbol{e}_j$ is the measurement noise of the $j$-th coil, $N$ is the number of coils, $\mathbf{C}_j$ denotes a diagonal matrix representing the $j$-th coil sensitivity map, $\mathcal{F}$ is the Fourier transform matrix, and $\mathbf{M}$ is the sampling mask. For simplicity, we omit the coil subscript $j$ in the following derivations. Defining $\mathcal{A} = \mathbf{M}\mathcal{F}\mathbf{C}$ as the physical forward model, the inverse problem can be formulated as:

$$\boldsymbol{I}^* = \arg\min_{\boldsymbol{I}} \quad \frac{1}{2}\|\boldsymbol{S} - \mathcal{A}\boldsymbol{I}\|_2^2 + \lambda \cdot \|\boldsymbol{G}\boldsymbol{I}\|_1, \quad (2)$$

where $\boldsymbol{G}$ denotes the gradient operator enforcing smoothness regularization, and $\lambda$ is a hyper-parameter controlling its contribution. The key to addressing this ill-posed problem is the design of a reliable prior that effectively constrains the solution space, thus enabling desired image reconstructions. Since the explicit total variation regularizer cannot fully capture the complex distribution of MR images, its performance remains limited.

**Conventional INR for MRI Reconstruction** Implicit neural representation (INR) formulates the complex-valued MR image $\boldsymbol{I}$ as a continuous function of spatial coordinates (Feng et al., 2022; 2023), which are parametrized by

two separate MLP networks as follows:

$$I(p) = f_{\text{real}}(p) + i \cdot f_{\text{imag}}(p), \quad (3)$$

where $p = (x, y)$ denotes 2D spatial coordinates in the normalized imaging space $\Omega = [-1, 1] \times [-1, 1]$. The functions $f_{\text{real}} : \mathbb{R}^2 \to \mathbb{R}$ and $f_{\text{imag}} : \mathbb{R}^2 \to \mathbb{R}$ correspond to MLPs representing the real and imaginary components, respectively, with $\theta$ and $\phi$ as their learnable parameters.

As illustrated in Fig. 1, INR optimizes the two networks $f_{\text{real}}$ and $f_{\text{imag}}$ to recover high-quality MR images in a self-supervised manner. Specifically, the networks take all coordinates $p \in \Omega$ as input and predict the corresponding real part $f_{\text{real}}(p)$ and imaginary part $f_{\text{imag}}(p)$. The predicted image $I$ is then transformed into $k$-space estimations $\mathcal{A}I$. Since the forward model $\mathcal{A}$ is differentiable, the networks can be optimized using gradient-based backpropagation. Consequently, $f_{\text{real}}$ and $f_{\text{imag}}$ are jointly trained by minimizing the loss function $\mathcal{L}$, which consists of a data consistency term and a smoothness regularizer. Formally, INR solves the following optimization:

$$\Theta^* = \arg\min_{I} \quad \frac{1}{2}\|S - \mathcal{A}I\|_2^2 + \lambda \cdot \|GI\|_1,$$
$$\text{subject to} \quad I(p) = f_{\text{real}}(p) + i \cdot f_{\text{imag}}(p). \quad (4)$$

where $\Theta^* = (\theta^*, \phi^*)$ denotes the optimal parameters. Benefiting from the inherent inductive bias of neural networks toward continuous image structures, i.e., spectral bias (Rahaman et al., 2019), the INR parameterization enables accurate approximation of the underlying continuous function. After optimization, feeding all coordinates $p$ into the trained networks $f_{\text{real}}^*$ and $f_{\text{imag}}^*$ yields the final high-quality MR image reconstruction $I^*(p) = f_{\text{real}}^*(p) + i \cdot f_{\text{imag}}^*(p)$.

**Our Motivation**   Although existing INR-based methods for the MRI reconstruction task demonstrate great potential, their scan-specific optimization faces two key limitations: 1) they fail to exploit population-level data priors, limiting performance for diverse cases; 2) they start optimization from randomly initialized networks, reducing convergence stability and speed.

To this end, we aim to learn effective initializations for INR networks, thereby enabling rapid and robust adaptation to diverse MRI scenarios. Specifically, we propose an inverse-problem-driven framework, namely IPOD, which can learn generalized and robust INR initializations directly from population-level undersampled $k$-space data *without* the need for any high-quality MR images. Our key insight is that diverse MRI inverse problems share underlying similarities in their solution manifolds. By learning from a population of inverse problems during meta-training, IPOD captures common patterns and embeds them into initialization parameters. These population-level priors provide an informed starting point, enabling initialized INRs to generalize effectively across different imaging scenarios with varying anatomy, contrast mechanisms, and sampling patterns.

### 3.2. Inverse problem-driven Meta-Learning

Fig. 2 (a) shows the pipeline of INR initializations by the proposed IPOD. Technically, IPOD adapts the Reptile (Nichol et al., 2018) algorithm to meta-initialize both $f_{\text{real}}$ and $f_{\text{imag}}$ across multiple inverse reconstruction problems, eliminating the need for fully-sampled $k$-space data while leveraging physics-based forward models. It consists of three main components, which we detail in the following sections.

**Inverse Problem Set Construction**   To enable effective meta-learning across diverse MRI reconstruction scenarios, we construct a comprehensive inverse problems set $\mathcal{S} = \{S_1, S_2, \ldots, S_L\}$ comprising various undersampled $k$-space data. To ensure balanced representation and mitigate bias, we systematically curate the task set to include different undersampling ratios, various $k$-space trajectories, and multiple image categories. This diversity enables our framework to learn generalizable initialization strategies that are robust across different MRI applications.

**Problem-Specific Inner Loop**   Within each inner loop iteration, an undersampled $k$-space data is first sampled from the task set, i.e., $S_n \in \mathcal{S}$, and the INR-based reconstruction process is conducted to solve the corresponding inverse problem like the optimization procedure in Fig. 1. Specifically, given the sampled data $S_n$, we use the corresponding INR network $f_{\text{real}}^n$ and $f_{\text{imag}}^n$ with learnable parameter $\Theta_n$ to predict the underlying MR image $I_n(p) = f_{\text{real}}(p) + i \cdot f_{\text{imag}}(p)$. Then, we use the physical forward model $\mathcal{A}_n = M_n \mathcal{F} C_n$ specific to the measurement $S_n$ for generating the $k$-space estimate $\mathcal{A}_n I_n$. Finally, we calculate the problem-specific loss $\mathcal{L}_n$, which is calculated as:

$$\mathcal{L}_n(\Theta_n, S_n) = \frac{1}{2}\|S_n - \mathcal{A}_n I_n\|_2^2 + \lambda \cdot \|GI_n\|_1. \quad (5)$$

These parameters $\Theta_n$ are iteratively optimized via gradient descent algorithms to minimize the physics-informed loss function, with updates computed as:

$$\Theta_n \leftarrow \Theta_n - \gamma \nabla_{\Theta_n} \mathcal{L}_n(\Theta_n, S_n), \quad (6)$$

where $\gamma$ denotes the inner loop learning rate. This optimization process adapts the INR parameters to the specific characteristics of each inverse problem, and the resulting problem-adapted parameters $\Theta_n$ are subsequently utilized in the meta-update phase.

**Adaptive Weighted Meta-optimization**   In each meta-learning epoch, we first randomly sample a subset from

*Table 1.* Quantitative results (Mean $\pm$ STD in PSNR) comparing different INR backbones with and without proposed **IPOD** initialization on different datasets, anatomies, sampling patterns and physical forward models. The best performance for each method type is highlighted in **bold**.

| Dataset | Sampling | AF | DINER | | SIREN | | HASH | |
|---|---|---|---|---|---|---|---|---|
| | | | w/o IPOD | w/ IPOD | w/o IPOD | w/ IPOD | w/o IPOD | w/ IPOD |
| Brain (fastMRI) | Cartesian | 3 | 58.58±2.92 | **58.98±3.14** | 44.50±3.97 | **55.24±5.34** | **60.68±4.91** | 59.88±4.41 |
| | | 4 | 48.61±5.12 | **48.89±4.87** | 37.12±3.62 | **41.61±5.16** | 47.32±3.37 | **47.78±3.25** |
| | Radial | 10 | 36.73±3.79 | **39.69±1.66** | 32.73±3.00 | **37.23±3.03** | 38.95±2.44 | **39.26±2.12** |
| | | 14 | 33.41±4.08 | **36.87±1.78** | 31.08±2.42 | **34.94±1.65** | 35.99±2.04 | **36.21±2.42** |
| Knee (fastMRI) | Cartesian | 3 | 51.20±5.11 | **54.24±4.01** | 36.06±2.07 | **43.45±3.95** | 54.21±5.87 | **56.63±2.16** |
| | | 4 | 39.25±3.95 | **40.33±2.32** | 35.96±1.71 | **38.86±1.81** | 39.31±2.30 | **40.50±1.89** |
| | Radial | 10 | 33.76±2.66 | **35.14±1.60** | 30.13±2.30 | **33.30±1.50** | 34.94±1.62 | **34.96±1.61** |
| | | 14 | 31.25±3.21 | **33.46±1.48** | 29.21±1.79 | **31.26±1.50** | 33.03±1.71 | **33.17±1.61** |
| Brain (MoDL) | Cartesian | 3 | 49.80±4.18 | **53.93±4.51** | 52.24±1.39 | **59.18±1.89** | 57.60±3.09 | **59.36±2.65** |
| | | 4 | 40.08±3.55 | **52.13±4.62** | 41.77±1.11 | **53.69±1.55** | 56.71±3.21 | **58.07±2.40** |
| | Radial | 10 | 32.15±2.04 | **37.76±2.68** | 30.31±0.52 | **34.46±3.60** | 37.54±1.51 | **37.72±0.78** |
| | | 14 | 29.46±2.13 | **35.75±0.49** | 29.88±0.61 | **32.56±0.59** | 34.69±0.80 | **35.00±1.18** |

the task set $\mathcal{S}_{\text{sub}} = \{\boldsymbol{S}_1, \boldsymbol{S}_2, \ldots, \boldsymbol{S}_N\} \subseteq \mathcal{S}$. For each task $\boldsymbol{S}_n \in \mathcal{S}_{\text{sub}}$, the problem-adapted parameters $\boldsymbol{\Theta}_n$ are updated through inner-loop optimization (Eq. 6). After completing all inner loops, we obtain a parameter space $\mathcal{P} = \{(\boldsymbol{\Theta}_1, \mathcal{L}_1), (\boldsymbol{\Theta}_2, \mathcal{L}_2), \ldots, (\boldsymbol{\Theta}_N, \mathcal{L}_N)\}$, where each pair represents problem-specific parameters and the corresponding reconstruction loss. However, severely ill-posed inverse problems may lead to suboptimal parameter learning due to insufficient constraints and ambiguous solution spaces. Specifically, when confronted with challenging reconstructions, the optimization may converge to poor local minima or exhibit unstable convergence behavior, resulting in network parameters that inadequately represent the underlying image structure. Directly incorporating such parameters into meta-updates can degrade the overall initialization.

To alleviate such potential negative effect, we introduce an adaptive weighting strategy that selectively emphasizes inverse problems with better solution space. Specifically, each task is assigned a weight based on the inverse of its loss:

$$\boldsymbol{w_n} = \frac{1/(\mathcal{L}_n + \epsilon)}{\sum_{n=1}^{N} 1/(\mathcal{L}_n + \epsilon)}, \tag{7}$$

where $\epsilon = 1 \times 10^{-8}$ prevents division by zero. The meta-parameters $\boldsymbol{\Theta}_m$ are then updated using a weighted variant of Reptile (Nichol et al., 2018):

$$\boldsymbol{\Theta}_m \leftarrow \boldsymbol{\Theta}_m + N\alpha \sum_{n=1}^{N} \boldsymbol{w_n}(\boldsymbol{\Theta}_n - \boldsymbol{\Theta}_m). \tag{8}$$

This mechanism reduces the impact of poorly-performing tasks and yields robust initializations that generalize across

diverse sampling patterns and anatomical structures. After all updates, we obtain the meta-initialized INR networks, which serve as efficient, robust, and broadly generalizable initializations, facilitating reliable adaptation to diverse undersampled MRI reconstruction scenarios.

### 3.3. Inference on Unseen Undersampled $k$-space Data

Once meta-training is completed, the meta-initialized parameters $\boldsymbol{\Theta}_m$ are employed as the starting point to finetune INR networks on unseen undersampled $k$-space data. As shown in Fig. 2 (b), the finetuning procedure follows the same adaptation scheme as the inner-loop updates during meta-training, where $\boldsymbol{\Theta}_m$ is refined through several optimization iterations. Importantly, the meta-initialized INRs are not restricted to the inverse problem categories seen during meta-training, but can effectively generalize to undersampled data drawn from broader and unseen data distributions, enabling robust reconstructions across diverse MRI scenarios.

## 4. Experiments

### 4.1. Experimental Settings

**Datasets** The retrospective experiments are conducted on fastMRI (Knoll et al., 2020) and MoDL datasets (Aggarwal et al., 2018). For meta-training, we extract 3,600 undersampled $k$-space measurements from fastMRI, encompassing different contrasts (T2w, FLAIR), anatomical structures (brain, knee) with $256 \times 256$ resolution. For evaluation, we test on unseen scenarios including different subjects and contrasts. Specifically, we use 50 brain slices (T1w,

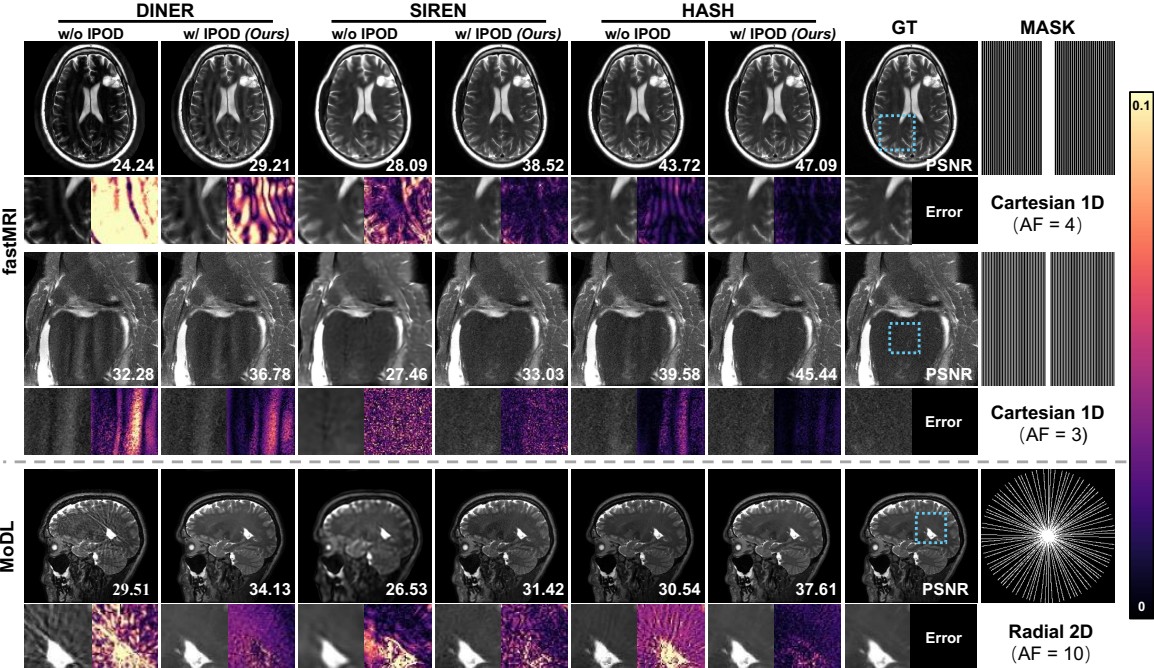

*Figure 3.* Qualitative and quantitative comparison of baselines with and without IPOD initialization under a small number of parameter updates (150 iterations) on fastMRI and MoDL datasets.

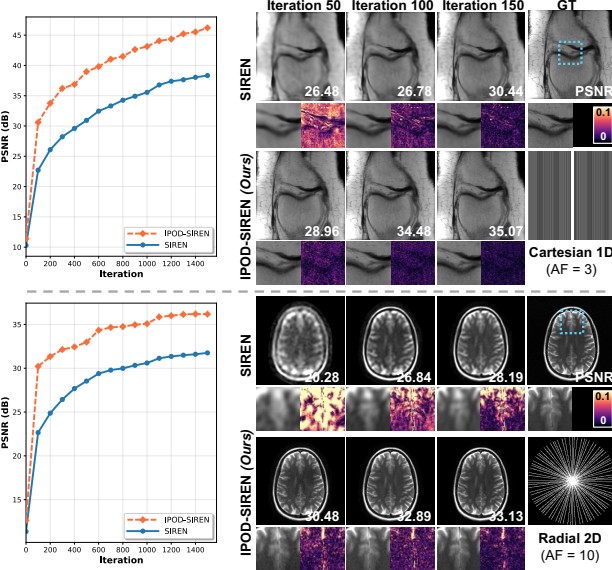

*Figure 4.* Comparison of performance curves and qualitative/quantitative results of SIREN with and without IPOD initialization under varying parameter update iterations on different datasets.

T2w, FLAIR) and 50 knee slices from the fastMRI dataset, plus 50 T2w brain slices with $256 \times 256$ from 20 subjects in the MoDL dataset. Additionally, we acquired two distinct 3D T1w brain datasets using a 3.0 T United Imaging Healthcare (UIH) uMR 890 scanner with a 3D MPRAGE

sequence: (1) a prospective T1w scan for real-world validation, and (2) retrospective 3D scans from five subjects for high-resolution reconstruction benchmarking. These 3D data were processed via 1D-IFT along the slice-encoding direction and subsequently undersampled using various masks and acceleration factors.

**Pre-processing** For meta-learning training data, we employ two distinct sampling patterns: 1D Cartesian undersampling and 2D random undersampling. The 1D Cartesian sampling encompasses acceleration factors (AFs) of 2, 4, and 6, while the 2D random sampling includes AFs of 5, 10, and 15. To validate the generalization of initialization parameters, we additionally implement three unseen sampling patterns: 2D Poisson sampling, 2D Gaussian sampling, and 2D radial sampling with golden-angle acquisition scheme. *More details about pre-processing can be found in the Appendix.*

**Baselines and Metrics** We select three representative INR architectures as backbones: DINER (Xie et al., 2023), SIREN (Sitzmann et al., 2020b), and HASH (Müller et al., 2022), covering methods with non-linear activation functions and MLP structures with coordinate encoders. To validate IPOD's universal applicability, we apply our meta-learned initialization to each backbone, creating IPOD-DINER, IPOD-SIREN, and IPOD-HASH variants, and compare against their randomly initialized counterparts. Metrics (PSNR and SSIM) are computed within regions of inter-

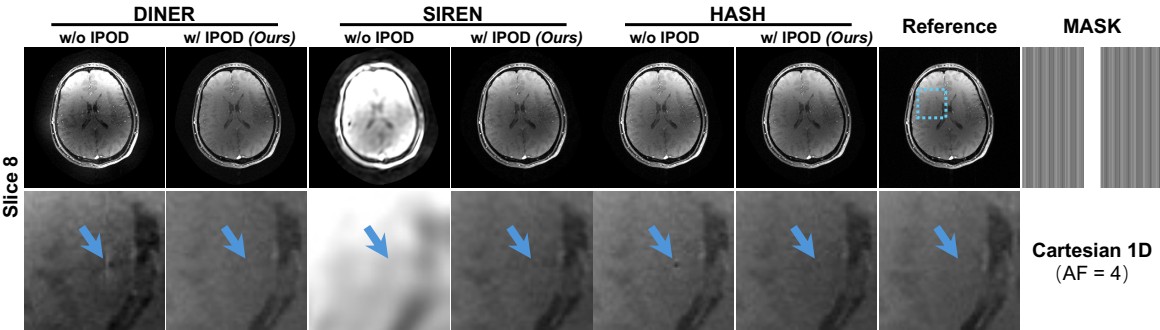

*Figure 5.* Qualitative comparison of baselines with and without IPOD initialization under a small number of parameter updates (75 iterations) on prospectively undersampled real-world data.

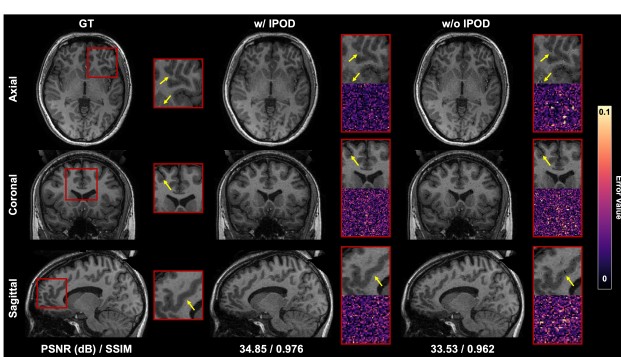

*Figure 6.* Qualitative and quantitative comparison of 3D reconstruction results with and without IPOD (HASH) under Poisson undersampling (AF = 6). Yellow arrows indicate the abnormal structures in results without IPOD.

*Table 2.* Quantitative comparison (Mean ± Std in PSNR/SSIM) of 3D reconstruction results with and without IPOD across different acceleration factors and sampling patterns.

| Sampling Pattern | AF | Metric | w/ IPOD | w/o IPOD |
|---|---|---|---|---|
| 1D Cartesian | 4 | PSNR | **35.21 ± 2.09** | 34.33 ± 1.96 |
|  |  | SSIM | **0.985 ± 0.008** | 0.971 ± 0.009 |
| 2D Poisson | 6 | PSNR | **33.21 ± 1.37** | 31.52 ± 2.09 |
|  |  | SSIM | **0.978 ± 0.007** | 0.964 ± 0.009 |
| 2D Gaussian | 8 | PSNR | **31.60 ± 1.49** | 29.78 ± 1.33 |
|  |  | SSIM | **0.963 ± 0.014** | 0.956 ± 0.013 |

est (ROI) to emphasize reconstruction quality of anatomical details. We further benchmark against five established methods—classical optimization-based (SENSE (Pruessmann et al., 1999), GRAPPA (Griswold et al., 2002)), self-supervised (ConvDecoder (Darestani & Heckel, 2021), SSDU (Yaman et al., 2020)), and supervised (E2E-VN (Sriram et al., 2020))—with full-image region metrics to maintain consistency with their original evaluation protocols. *More experiment details and results can be found in the Appendix.*

**Implementation Details**  All methods are implemented on PyTorch. For the IPOD meta-learning phase, we construct the set $\mathcal{S}$ of inverse problems with 72 different types, where each type contains 50 samples from different subjects, resulting in $L = 3,600$ total undersampled $k$-space data. We set the batch size $N$ in each meta-learning epoch to 15, and the number of meta-learning training epochs to 2,500 with a meta-learning rate $\alpha = 5 \times 10^{-4}$. For inner loop optimization, we set different learning rates for each backbone: $\gamma = 2 \times 10^{-2}$ for DINER and HASH, $\gamma = 2 \times 10^{-4}$ for SIREN during meta-learning. The total variation regularization weight is fixed to $\lambda = 2$ across all datasets. We use the

Adam algorithm with default hyper-parameters (Kingma & Ba, 2014) to optimize all the models. Notably, two types of physical forward models are employed: the fast Fourier transform (FFT) and Non-uniform Fast Fourier Transform (NuFFT) (Fessler, 2007). All experiments were performed on a single NVIDIA RTX A100 GPU. *Due to the space constraint, we introduce the other implementation details of IPOD and baselines in the Appendix.*

### 4.2. Main Results

**Performance-Oriented Comparison with Baselines**  Table 1 presents quantitative PSNR results for different INR architectures with and without IPOD initialization. IPOD consistently improves reconstruction quality across all backbones, datasets, and sampling patterns, with particularly pronounced gains under challenging conditions such as higher acceleration factors and radial sampling. Among all architectures, SIREN benefits most from IPOD initialization.

**Efficiency-Oriented Comparison with Baselines**  To demonstrate the fast generalization ability of IPOD-initialized networks on unseen data, we conducted an efficiency comparison with baselines using only 150 parameter updates. As shown in Fig. 3, IPOD consistently achieves significant performance improvements over randomly initialized baselines. The error maps further validate our approach, showing reduced reconstruction errors and fewer artifacts

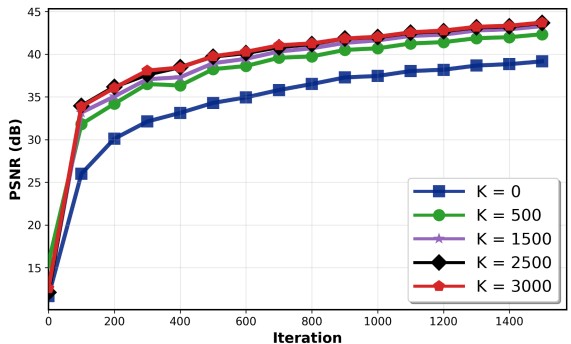

*Figure 7.* The performance and convergence comparison between backbone SIREN (K = 0) and IPOD-SIREN with different meta-learning training epochs (K) on unseen samples from the fastMRI brain dataset for AF = 4.

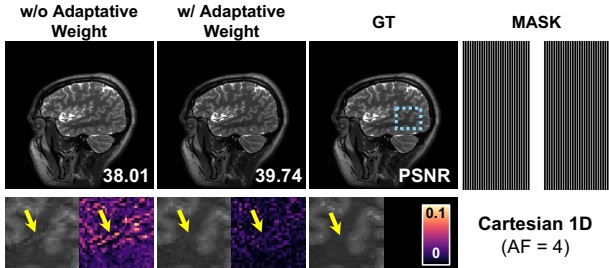

*Figure 8.* Qualitative and quantitative comparison showing the influence of the adaptive weight mechanism in the IPOD framework. Results are from IPOD-SIREN on a sample from the MoDL dataset.

*Table 3.* Quantitative evaluation showing the influence of the adaptive weight mechanism in the IPOD framework on reconstruction performance. Results are from IPOD-SIREN on the MoDL dataset for AF = 4.

| Method | PSNR (dB) | SSIM |
|---|---|---|
| w/o Adaptive Weight | 36.38±2.20 | 0.978±0.025 |
| w/ Adaptive Weight | **37.14±2.41** | **0.980±0.023** |

compared to baselines.

**Convergence Evolution Analysis against SIREN Baseline**
As depicted in Fig. 4, we analyze the convergence of IPOD-initialized SIREN against randomly initialized baseline. The performance curves demonstrate that IPOD achieves significantly faster convergence and superior final performance. The reconstructed images further validate this efficiency, with IPOD-SIREN producing notably better results at early iterations (50 and 100) for both Cartesian 1D and radial 2D sampling. Error maps reveal that IPOD learns to remove reconstruction artifacts much faster, achieving superior image quality and PSNR scores by 150 iterations.

*Table 4.* Average GPU reconstruction time and cost comparison across optimization-based methods, deep learning methods, and IPOD with different backbones.

| Method | Time (w/o IPOD) | Time (w/ IPOD) | GPU Mem. (GB) |
|---|---|---|---|
| SENSE-L1 | 3.5 s | – | 0.83 |
| SENSE-TV | 15.3 s | – | 0.79 |
| GRAPPA | 4.8 s | – | 0.75 |
| E2E-VN[†] | 1.3 s / 16.4 h | – | 2.24 / 56.4 |
| ConvDecoder | 192.5 s | – | 1.80 |
| SSDU | 42.6 s | – | 8.26 |
| DINER[‡] | 13.5 s | 5.8 s / 3.1 h | 1.52 / 20.9 |
| SIREN[‡] | 116.7 s | 37.8 s / 4.5 h | 1.93 / 24.2 |
| HASH[‡] | 11.6 s | 4.2 s / 3.1 h | 2.16 / 21.3 |

[†] For E2E-VN, time values denote Inference / Training; memory values denote Inference / Training respectively. [‡] For INR methods with IPOD, time values denote Reconstruction / Meta-Learning (ML); memory values denote Inference / ML respectively.

**Reconstruction Performance on 3D MRI Data**    As illustrated in Fig. 6 and Table 2, IPOD consistently outperforms the baseline (HASH w/o IPOD) across various acceleration factors and sampling patterns. Benefiting from the learned robust prior, IPOD preserves superior structural details and effectively suppresses noise interference, especially in challenging high-resolution acquisitions.

**Reconstruction Results for Real-World Data**    As shown in Fig. 5, we evaluated baselines with and without IPOD initialization on real-world prospectively undersampled data. Notably, the T1w modality was unseen during meta-learning training, yet IPOD remained effective with only 75 parameter updates. IPOD-initialized models achieve superior reconstruction quality, producing sharper images with reduced artifacts. This is particularly evident in magnified regions, where IPOD successfully recovers structures that exhibit distortions in baseline reconstructions, validating IPOD's strong generalization capability to real-world measurements.

**Reconstruction Time and Cost Comparisons**    Table 4 summarizes the reconstruction time and GPU memory consumption across optimization-based methods, deep learning methods, and IPOD with different backbones. To ensure a fair comparison, all methods except E2E-VN utilized a consistent convergence criterion (PSNR change rate $< 0.01$ dB/iteration). IPOD-initialized models consistently achieve a $2\times$ to $3\times$ reconstruction speedup across all backbones. Compared to the supervised E2E-VN, IPOD avoids the substantial training overhead and the dependency on fully-sampled data, while achieving competitive reconstruction speed at inference.

*Table 5.* Ablation on problem set diversity: quantitative comparison of SIREN backbone across progressive iterations on radial undersampling.

| Method | Iter 10 | | Iter 100 | | Iter 500 | | Iter 1000 | |
|---|---|---|---|---|---|---|---|---|
| | PSNR | SSIM | PSNR | SSIM | PSNR | SSIM | PSNR | SSIM |
| w/o IPOD | 12.74±0.98 | 0.510±0.056 | 23.90±1.87 | 0.772±0.006 | 29.17±0.31 | 0.891±0.017 | 31.47±0.45 | 0.934±0.006 |
| IPOD-Cartesian | 16.40±4.36 | 0.530±0.017 | 26.80±0.84 | 0.832±0.011 | 31.91±0.25 | 0.927±0.002 | 32.92±0.79 | 0.953±0.008 |
| IPOD-All | **18.51±1.00** | **0.604±0.022** | **30.97±0.99** | **0.907±0.009** | **33.78±1.76** | **0.957±0.007** | **34.21±2.96** | **0.968±0.006** |

*Table 6.* Ablation on problem set diversity: quantitative comparison of SIREN backbone across progressive iterations on fastMRI knee dataset.

| Method | Iter 10 | | Iter 100 | | Iter 500 | | Iter 1000 | |
|---|---|---|---|---|---|---|---|---|
| | PSNR | SSIM | PSNR | SSIM | PSNR | SSIM | PSNR | SSIM |
| w/o IPOD | 13.81±2.62 | 0.539±0.057 | 26.59±3.11 | 0.833±0.050 | 29.99±2.81 | 0.904±0.026 | 31.71±2.83 | 0.929±0.018 |
| IPOD-Brain | 17.74±1.08 | 0.667±0.042 | 29.33±3.53 | 0.899±0.040 | 31.91±3.68 | 0.937±0.028 | 33.14±3.38 | 0.950±0.023 |
| IPOD-All | **19.57±2.58** | **0.726±0.044** | **30.72±3.35** | **0.919±0.029** | **33.04±3.93** | **0.951±0.020** | **34.74±3.58** | **0.960±0.017** |

## 4.3. Ablation Studies

**Influence of Meta-learning epochs**   Fig. 7 illustrates the performance of the SIREN network as a function of meta-learning epochs (K). The curves show that even a small number of epochs (K = 500) significantly boosts the performance of our IPOD-initialized SIREN networks over its randomly initialized counterpart (K = 0). As K increases, the effectiveness of initialization steadily improves, leading to faster convergence and higher final PSNR scores before reaching a plateau. This shows that our IPOD framework enables the baseline model to learn a more robust initialization for MRI reconstruction.

**Influence of Adaptive Weight**   Table 3 demonstrates the effectiveness of the adaptive weight mechanism for efficient initialization within our IPOD framework, showing a significant drop in reconstruction performance when this component is removed. This effect is further illustrated in Fig. 8, where the model trained without adaptive weights produces noticeable artifacts in the final reconstructed images. This clearly indicates that without this mechanism, the meta-learning process incorporates suboptimal parameter distributions, which negatively impacts the final reconstruction quality.

**Influence of Dataset Diversity**   To evaluate the effect of problem set diversity, we compare three meta-learning strategies: (1) IPOD-Brain, trained only on brain anatomy; (2) IPOD-Cartesian, trained only on Cartesian undersampling; and (3) IPOD-All, trained on the full diverse problem set. All hyperparameters remain consistent. As shown in Tables 5 and 6, IPOD-All consistently outperforms the specialized variants across all iterations, demonstrating that a diverse problem set yields more robust and generalizable priors. Notably, IPOD-Cartesian still improves over random

initialization on unseen radial data, but the gap to IPOD-All highlights the benefit of sampling pattern diversity. *Qualitative comparisons can be found in the Appendix.*

## 5. Limitations

While IPOD demonstrates strong generalization across diverse MRI reconstruction scenarios, some directions remain open. (1) IPOD currently uses only $k$-space measurements; incorporating textual priors (e.g., imaging protocol metadata) could yield protocol-aware initialization for multimodal reconstruction. (2) Our evaluation is limited to static brain and knee data; extension to broader and temporally-resolved modalities such as dynamic MRI and 4D flow requires handling additional temporal dimensions and motion. (3) The current 3D pipeline reconstructs slice-by-slice, discarding potential inter-slice correlations; a native volumetric meta-initialization would better exploit full 3D spatial coherence.

## 6. Conclusion

We presented IPOD, an inverse-problem-driven meta-learning framework that achieves robust INR initialization without requiring fully-sampled reference data. By integrating physics-informed meta-optimization with an adaptive weighting mechanism, IPOD overcomes the data-dependency constraints and domain-shift challenges inherent in conventional medical imaging. Extensive experimental results on both simulated and prospective clinical datasets demonstrate that IPOD consistently achieves superior reconstruction fidelity and faster convergence across diverse anatomical structures, contrast mechanisms, and complex sampling patterns from various scanners.

## Acknowledgements

The study is supported by the National Key Research and Development Program of China (2024YFC2421100), the Shanghai Science and Technology Development Funds, China (23TS1400200), the National Natural Science Foundation of China (62471296) and the SJTU Trans-med Awards Research (STAR 20220103, YG2023LC02).

## Impact Statement

This paper presents work whose goal is to advance the field of Machine Learning. There are many potential societal consequences of our work, none which we feel must be specifically highlighted here.

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

# A. Appendix

## A.1. Additional Details of Pre-processing

In the validation experiments, the generation of 2D Gaussian and Poisson undersampling masks followed the methodology described in (Chung & Ye, 2022).

## A.2. Additional Details of Metrics and Implementation

**Metrics for MR Image Quality** In our evaluation, two commonly used visual metrics—PSNR and SSIM—are employed to assess the quality of reconstructed MR images. These metrics are implemented using the Python library skimage (**https://github.com/scikit-image/scikit-image**).

**Implementation for Network Structures and Testing** The network architectures for $f_{real}$ and $f_{imag}$ vary across methods. For DINER, both networks use MLPs with 2 hidden layers of 16 neurons each, employing ReLU activation functions. SIREN networks consist of MLPs with 6 hidden layers of 256 neurons each, utilizing sinusoidal activation functions as specified in the original work. For HASH, both networks combine a hash-encoding module with an MLP containing 2 hidden layers of 16 neurons each with ReLU activations. We adopt the hyperparameters suggested in the original paper (Müller et al., 2022) for the hash-encoder configuration.

In the testing-time evaluation, all methods employ the AdamW optimizer with a fixed weight decay of $2 \times 10^{-4}$. For the DINER method, the initial learning rate is set to $1 \times 10^{-2}$ with a decay factor of 0.6 every 100 steps. For the SIREN method, the initial learning rate is $1 \times 10^{-4}$, where the baseline decays by 0.8 every 500 steps while the IPOD-initialized version decays by 0.8 every 100 steps. For the HASH method, the hash encoder uses a learning rate of $5 \times 10^{-2}$ and the MLP parameters use $1 \times 10^{-3}$, with both components decaying by 0.8 every 100 steps.

## A.3. Additional Details of Backbones

**DINER** Disorder-Invariant Implicit Neural Representation (DINER), is a framework that largely solves this frequency-related issue by re-arranging the coordinates of the input signal (Xie et al., 2023). DINER augments a traditional INR backbone with a learnable hash-table. This innovation allows the framework to handle discrete signals that share the same attribute histogram but have different spatial arrangements. The hash-table maps the input coordinates into a consistent distribution, regardless of the original arrangement order. This transformation results in a low-frequency mapped signal that can be better modeled by the subsequent INR network, leading to a significantly alleviated spectral bias. In this work, we reproduce the

MLP-based DINER structures based on their official code (**https://github.com/Ezio77/DINER**).

**SIREN** Implicit Neural Representation is a powerful paradigm, but its effectiveness is limited by its inability to accurately model signals with fine detail and spatial/temporal derivatives. Sinusoidal Representation Network (SIREN) leverages periodic activation functions to overcome these limitations (Sitzmann et al., 2020b). SIREN is ideally suited for representing complex natural signals and their derivatives. SIREN has been proven to be a superior self-supervised INR framework for inverse problems in MRI reconstruction (Feng et al., 2023). In this work, we reproduce the SIREN structures based on their official code (**https://github.com/vsitzmann/siren**).

**HASH** Here, HASH refers to the Instant Neural Graphics Primitives with a Multiresolution Hash Encoding (Instant-NGP) method (Müller et al., 2022). The method combines lightweight MLP networks with multi-resolution hash encoding, where coordinates are mapped through hash functions to feature vectors at different spatial resolutions. Multi-resolution hash encoding replaces traditional positional encoding with efficient hash tables at multiple resolution levels, significantly reducing computational overhead while maintaining high-quality representations. In this work, we reproduce the HASH structures based on their official code (**https://github.com/NVlabs/instant-ngp**).

## A.4. Additional Visual Results

Fig. 9, Fig. 10, Fig. 11, and Fig. 12 show additional reconstructed MR images through different baseline methods with and without IPOD initialization. The proposed IPOD initialization achieves fast adaptation across various imaging modalities and different inverse problems in MRI reconstruction.

## A.5. Additional Comparisons of Different Baselines

To provide a comprehensive evaluation, we benchmark IPOD against five established reconstruction methods: classical optimization-based (SENSE-TV, SENSE-L1Wavelet, GRAPPA), self-supervised (ConvDecoder, SSDU), and supervised (E2E-VN). We additionally report two perceptual metrics—LPIPS and DISTS—alongside PSNR and SSIM. All metrics are computed on the full image (without ROI cropping) to maintain consistency with the original evaluation protocols of these baselines.

**In-Domain Evaluation** Table 7 presents the quantitative comparison for 1D Cartesian undersampling (AF = 8) on the Brain dataset. IPOD-initialized backbones achieve competitive or superior performance compared to all self-supervised and optimization-based methods, and approach the super-

vised E2E-VN despite requiring no fully-sampled training data.

**Out-of-Distribution Evaluation**   To rigorously assess generalization, we designed a strict out-of-distribution (OOD) experiment. All methods are tested on an unseen Lesion Brain dataset with 2D Poisson undersampling (AF = 10). Crucially, the lesion data was unseen during both E2E-VN's supervised training and IPOD's meta-training, and the Poisson sampling pattern was also absent from both training pipelines.

As shown in Table 8, IPOD-initialized methods significantly outperform all baselines under this OOD setting. Notably, E2E-VN suffers severe performance degradation due to domain shift, while IPOD maintains robust reconstruction quality, demonstrating the strong generalization of physics-informed meta-learned priors.

### A.6. Ablation Study of Diversity in Problem Set

As described in the main text, we compare three meta-learning strategies (IPOD-Brain, IPOD-Cartesian, and IPOD-All) to evaluate the effect of problem set diversity. Quantitative results across progressive iterations are reported in Tables 5 and 6 of the main paper. Here we provide qualitative comparisons.

As shown in Figures 13 and 14, IPOD-Brain and IPOD-Cartesian at the early iteration stage (100 iterations) present artifacts and blurring of structural details, which are not observed in IPOD-All. This directly demonstrates that exposure to diverse inverse problems during meta-learning prevents prior overfitting to specific problem characteristics and produces more robust, generalizable initialization.

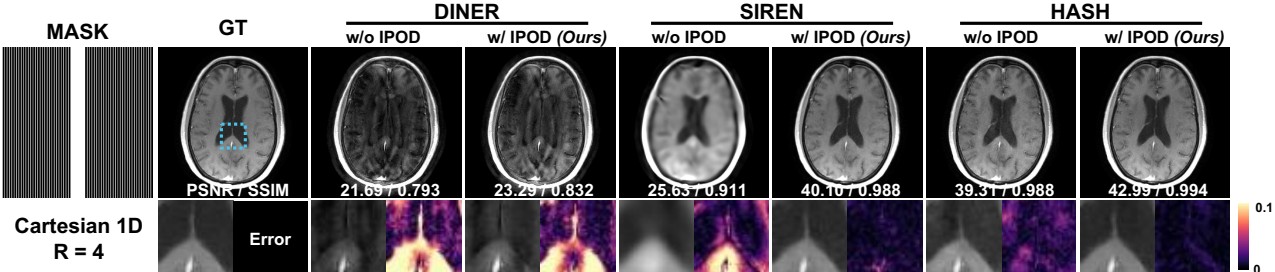

*Figure 9.* Qualitative and quantitative comparison of baselines with and without IPOD initialization under different parameter updates on fastMRI and MoDL datasets.

*Figure 10.* Qualitative and quantitative comparison of baselines with and without IPOD initialization under a small number of parameter updates (75 iterations) on the unseen T1w modality in fastMRI for AF = 4.

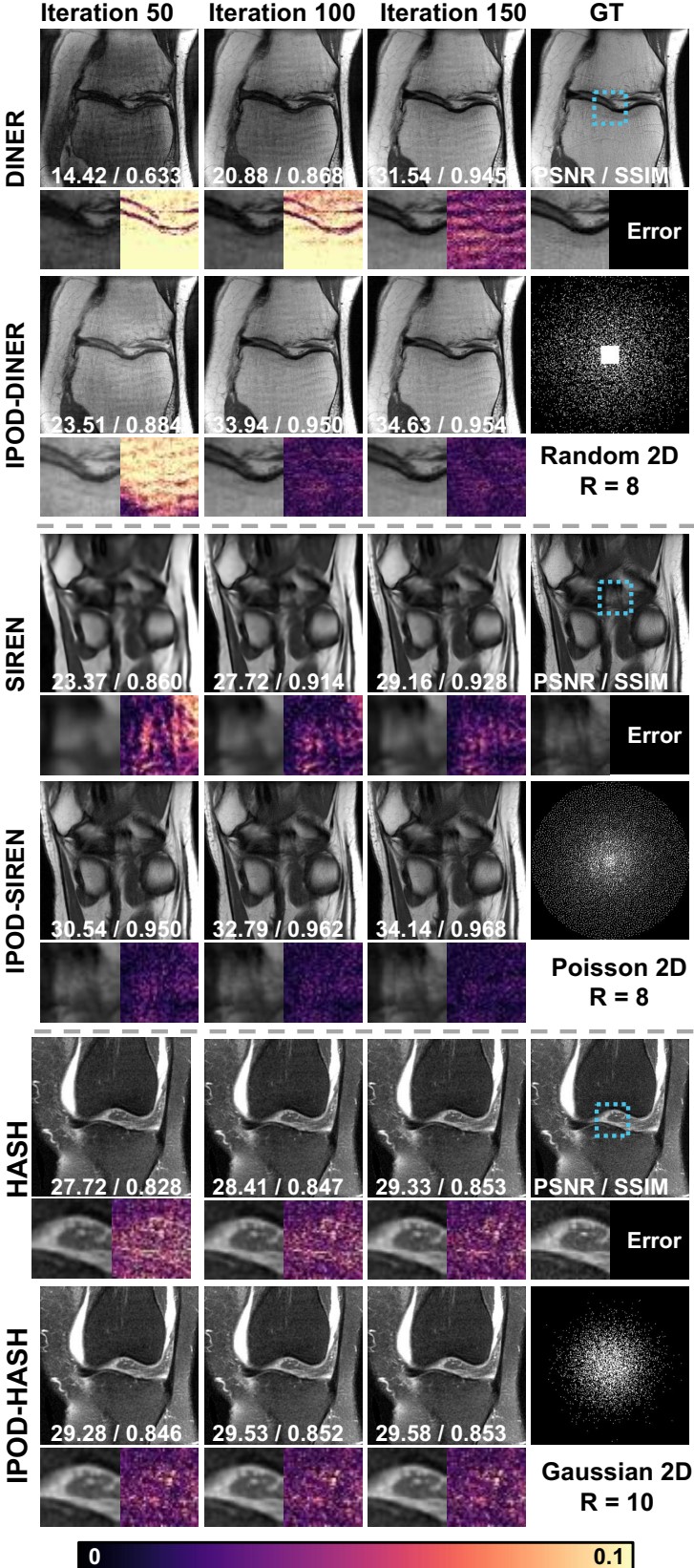

*Figure 11.* Qualitative and quantitative comparison of baselines with and without IPOD initialization under different parameter updates on fastMRI Knee datasets with different mask patterns.

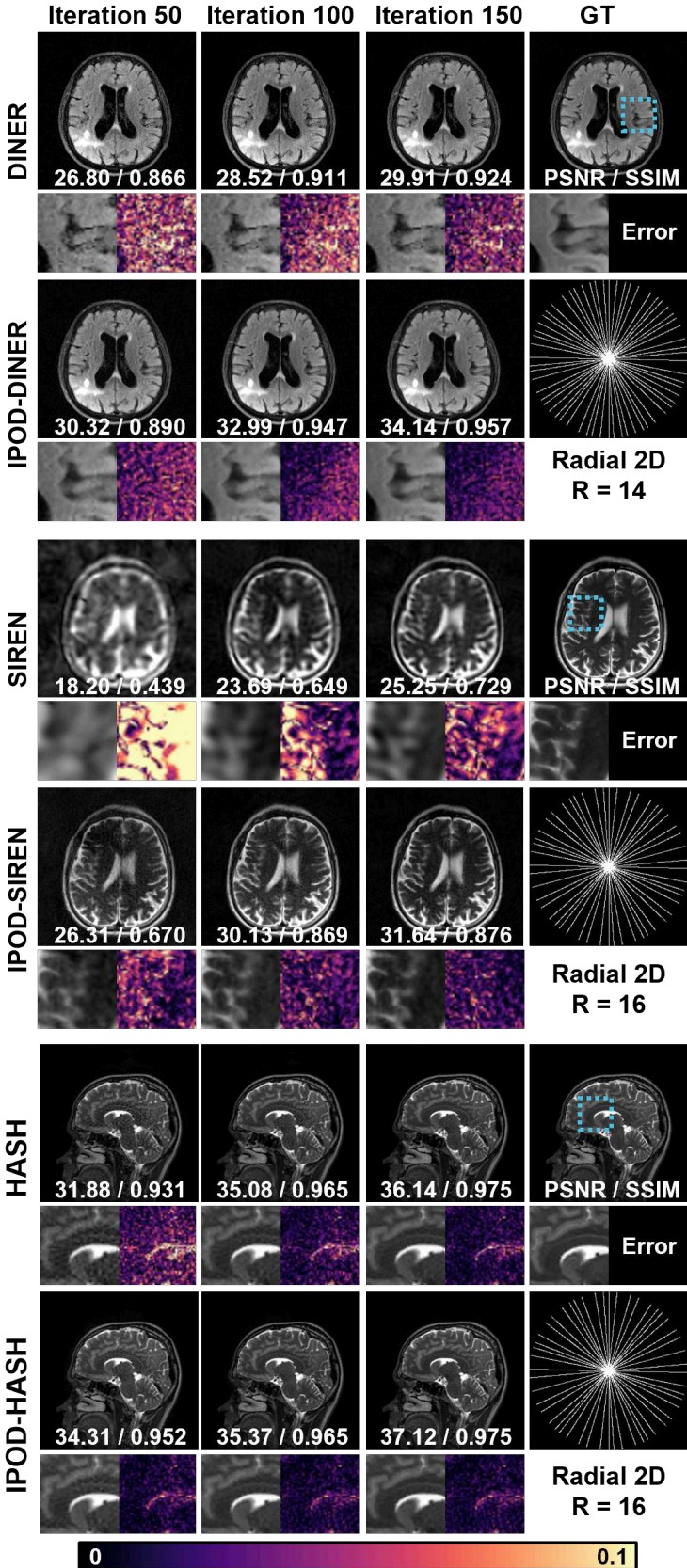

*Figure 12.* Qualitative and quantitative comparison of baselines with and without IPOD initialization under different parameter updates on fastMRI and MoDL datasets with different radial sampling patterns.

*Table 7.* Quantitative comparison of baselines and IPOD for 1D Cartesian undersampling (AF = 8) on Brain dataset. Full-image metrics.

| Method | PSNR (↑) | SSIM (↑) | LPIPS (↓) | DISTS (↓) |
|---|---|---|---|---|
| SENSE-TV | 25.32 ± 4.35 | 0.921 ± 0.011 | 0.1337 ± 0.0143 | 0.1890 ± 0.0138 |
| SENSE-L1Wavelet | 26.96 ± 1.24 | 0.924 ± 0.007 | 0.1390 ± 0.0022 | 0.1635 ± 0.0059 |
| GRAPPA | 25.07 ± 4.52 | 0.926 ± 0.014 | 0.1454 ± 0.0153 | 0.1699 ± 0.0156 |
| ConvDecoder | 32.43 ± 2.05 | 0.932 ± 0.013 | 0.0628 ± 0.0121 | 0.1385 ± 0.0112 |
| SSDU | 34.27 ± 1.78 | 0.935 ± 0.016 | 0.0486 ± 0.0104 | 0.1259 ± 0.0101 |
| E2E-VN (supervised) | **37.36 ± 1.46** | 0.939 ± 0.007 | **0.0224 ± 0.0068** | **0.1012 ± 0.0087** |
| DINER-IPOD | 34.23 ± 2.73 | 0.942 ± 0.018 | 0.0323 ± 0.0079 | 0.1180 ± 0.0135 |
| SIREN-IPOD | 35.47 ± 1.51 | 0.940 ± 0.012 | 0.0328 ± 0.0106 | 0.1201 ± 0.0116 |
| HASH-IPOD | 34.55 ± 2.86 | **0.944 ± 0.014** | 0.0318 ± 0.0081 | 0.1155 ± 0.0107 |

*Table 8.* OOD evaluation: quantitative results on unseen Lesion Brain dataset with 2D Poisson undersampling (AF = 10). Full-image metrics.

| Method | PSNR (↑) | SSIM (↑) | LPIPS (↓) | DISTS (↓) |
|---|---|---|---|---|
| SENSE-TV | 30.75 ± 1.07 | 0.916 ± 0.016 | 0.114 ± 0.012 | 0.108 ± 0.010 |
| SENSE-L1Wavelet | 30.18 ± 1.58 | 0.923 ± 0.023 | 0.087 ± 0.009 | 0.098 ± 0.011 |
| ConvDecoder | 32.75 ± 0.21 | 0.946 ± 0.004 | 0.062 ± 0.005 | 0.084 ± 0.006 |
| SSDU | 33.28 ± 0.49 | 0.950 ± 0.006 | 0.048 ± 0.004 | 0.072 ± 0.005 |
| E2E-VN (supervised) | 22.95 ± 1.00 | 0.788 ± 0.020 | 0.248 ± 0.010 | 0.216 ± 0.020 |
| DINER-IPOD | 33.58 ± 0.35 | 0.956 ± 0.002 | 0.021 ± 0.001 | 0.067 ± 0.003 |
| SIREN-IPOD | 33.85 ± 0.68 | **0.958 ± 0.002** | **0.019 ± 0.001** | **0.064 ± 0.003** |
| HASH-IPOD | **34.27 ± 0.45** | 0.957 ± 0.002 | 0.020 ± 0.001 | 0.066 ± 0.002 |

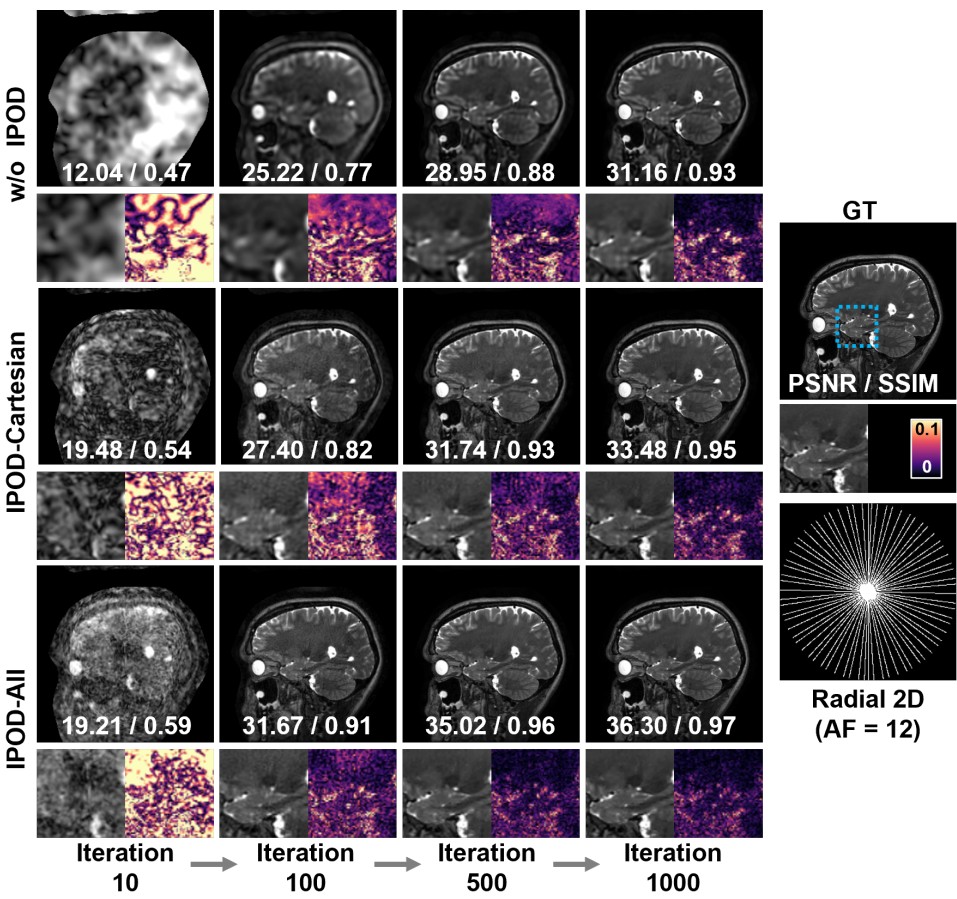

*Figure 13.* Qualitative and quantitative comparison of Baselines and IPOD with different meta-learning strategies across progressive iterations on MoDL dataset. (IPOD-Cartesian: meta-trained on Cartesian undersampling pattern only; IPOD-All: meta-trained on diverse undersampling patterns.)

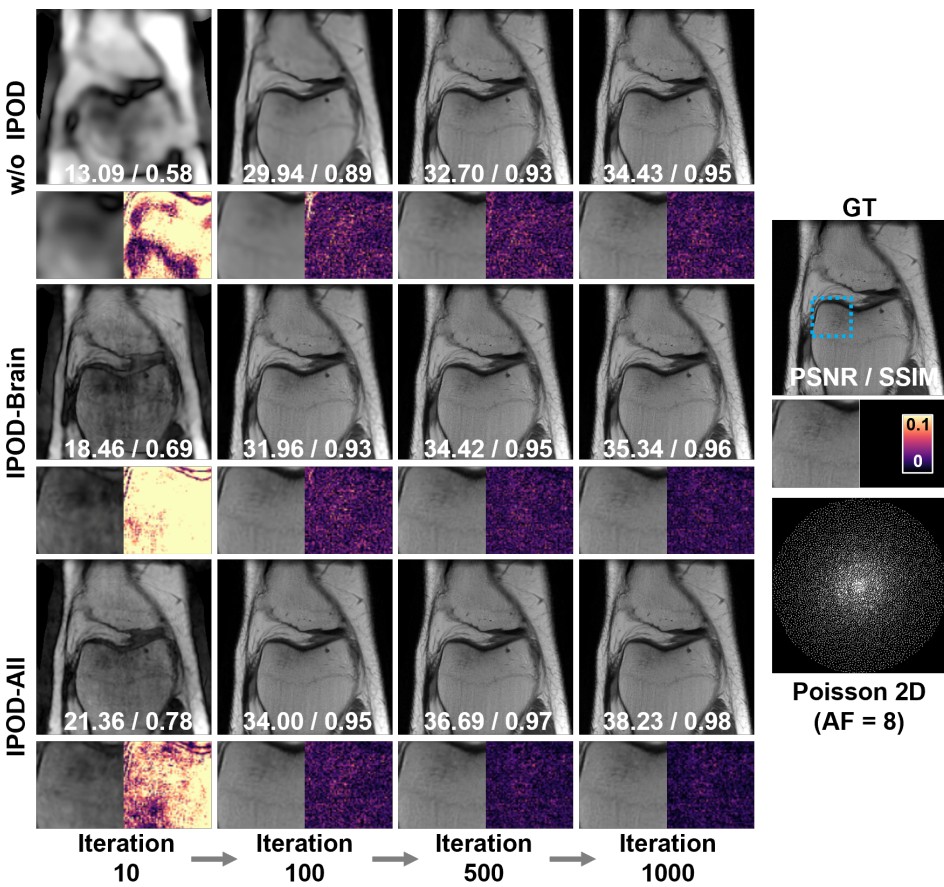

*Figure 14.* Qualitative and quantitative comparison of Baselines and IPOD with different meta-learning strategies across progressive iterations on fastMRI knee dataset. (IPOD-Brain: meta-trained on brain dataset only; IPOD-All: meta-trained on diverse anatomies dataset.)

