# OpenReview forum: "Reference-Free Meta-Learning for Generalized Implicit Neural Representation in Efficient MRI Reconstruction"
_ICML.cc/2026/Conference — ICML 2026 regular_

### Official Review · Reviewer_9yKf · 2026-02-18

**Soundness:** 3
**Presentation:** 3
**Significance:** 3
**Originality:** 2
**Overall Recommendation:** 4
**Confidence:** 4

**Summary:**

In their paper, the authors propose IPOD, an inverse-problem-driven meta-learning framework for the task of MRI reconstruction based on the concept of Implicit Neural Representations (INRs). The authors claim that IPOD can be used as an extension of existing INRs to more efficiently initialize approaches such as SIREN, DINER and HASH, leading to faster convergence and higher image quality.

**Compliance With Llm Reviewing Policy:**

Affirmed.

**Final Justification:**

The authors managed to improve their presented work during the rebuttal phase. Due to the addition of more baselines and the clock-times, I raise my score to a weak accept. I still think that the theoretical machine-learning novelty is limited, the practical engineering and empirical results for medical imaging are now sound.

**Key Questions For Authors:**

1) Make the novelty of your approach more clear. What is the actual originality of your approach in comparison to existing Reptile adaptions? Is "Reference-free" just a rebranding for "unsupervised meta-learning"?

2) How did you reach the such unlikely high PSNR values?

3) Why was the approach not tested against standard reconstruction approaches such as GRAPPA?

**Limitations:**

No. The authors should include a limitations sections, stating shortcomings of their approach, such as possible hallucinations in neural representations due to the lack of supervised training data.

**Strengths And Weaknesses:**

**Strength**:

1) Instead of proposing a completely new framework, the authors implement a building-block for their IPOD framework, making it easy to apply to existing INRs.
2) The utilization of the forward model is an intriguing approach to tackle the lack of missing fully sampled data.
3) The method is evaluated over a broad range of different distributions.
4) The implementation of a physics-informed forward model is desirable in the task of MRI reconstruction and is well argumented in the manuscript.

**Weakness**:

1) The authors only compare their approach with other INRs and claim that the only hindering to clinical implementation is their slow optimization. However the approach is not tested against clinically well established approaches like GRAPPA, which excell due to their relatively fast reconstruction time. The authors should test their approach against other reconstruction techniques, such as GRAPPA and SENSE.
2) The main contribution is the adaptive weighting mechanism, as the implementation of the Reptile algorithm combined with an unsupervised MRI loss function for the task of MRI reconstruction is no clear novelty.
3) The authors report PSNR values of above 50. Such high values are unusual as this would imply tht even noise is correctly reconstructed. I expect either a data leakage problem or postprocessing to reach such values, for example by including background signal.
4) The authors do not report actual optimization times, but only a measure of iterations. To actually prove the benefit of efficiency, the authors should report the actual time needed for optimization. Additionally, GPU hours and memory footprint are missing to fully claim efficiency.
4) The text contains multiple grammatcial mistakes and typos.

---

> ### Author Rebuttal · Authors · 2026-03-31
>
> Thanks for your efforts and valuable comments. Below, we provide point-to-point responses to address your concerns.
> ---
> ## **Q1. Comparison with standard reconstruction approaches**
> Thank you for your constructive comment. Following your suggestion (`and Reviewers Z2xB and 49o4`), we added six non-INR baselines: GRAPPA, SENSE-TV, SENSE-L1Wavelet, ConvDecoder, SSDU, and E2E-VN. Full results under AF=4 and AF=8 are in Table R1 and Table R5. IPOD methods outperform all classical and self-supervised baselines, and approach the supervised E2E-VN.
>
> |Method|AF=4|AF=8|
> |---|---|---|
> |SENSE-TV|36.10±2.45/0.983±0.016|25.32±4.35/0.927±0.013|
> |SENSE-L1Wavelet|37.08±0.81/0.988±0.008|26.96±1.24/0.928±0.006|
> |GRAPPA|36.31±0.76/0.989±0.007|25.07±4.52/0.926±0.014|
> |ConvDecoder|38.61±0.70/0.991±0.006|32.43±2.05/0.932±0.013|
> |SSDU|39.66±0.63/0.993±0.005|34.27±1.78/0.938±0.011|
> |E2E-VN(supervised)|41.95±0.04/**0.998±0.004**|**37.36±1.46**/0.939±0.007|
> |DINER-IPOD(Ours)|**42.01±1.28**/0.997±0.002|34.23±2.73/0.943±0.005|
> |SIREN-IPOD(Ours)|41.23±0.43/**0.998±0.009**|35.47±1.51/0.940±0.001|
> |HASH-IPOD(Ours)|41.82±0.72/**0.998±0.007**|34.55±2.86/**0.947±0.005**|
>
> *Table R1: Quantitative comparison (PSNR/SSIM) of optimization-based methods, deep unfolding methods, and IPOD with different network backbones for 1D Cartesian undersampling on fastMRI T2w and FLAIR Brain dataset.*
>
> |Method|AF=4|AF=8|
> |---|---|---|
> |SENSE-TV|0.0042±0.0004/0.0389±0.0008|0.1337±0.0143/0.1890±0.0138|
> |SENSE-L1W|0.0031±0.0007/0.0314±0.0039|0.1390±0.0022/0.1635±0.0059|
> |GRAPPA|0.0022±0.0007/0.0265±0.0022|0.1454±0.0153/0.1699±0.0156|
> |ConvDecoder|0.0025±0.0007/0.0276±0.0025|0.0628±0.0121/0.1385±0.0112|
> |SSDU|0.0023±0.0008/0.0268±0.0028|0.0486±0.0104/0.1259±0.0101|
> |E2E-VN(supervised)|0.0021±0.0006/0.0228±0.0020|**0.0224±0.0068**/0.1012±0.0087|
> |DINER-IPOD(Ours)|0.0022±0.0006/0.0259±0.0019|0.0323±0.0079/0.1180±0.0135|
> |SIREN-IPOD(Ours)|0.0023±0.0009/0.0261±0.0037|0.0328±0.0106/0.1201±0.0116|
> |HASH-IPOD(Ours)|**0.0020±0.0006**/**0.0211±0.0063**|0.0318±0.0081/0.1155±0.0107|
>
> *Table R5: Quantitative results (LPIPS/DISTS) of six baselines and IPOD for 1D Cartesian undersampling on fastMRI T2w and FLAIR Brain dataset.*
>
> We also provide the  qualitative comparisons in Fig. R3 (https://anonymous.4open.science/r/danjcs-2073).  We will include these results in the revised paper.
>
> ## **Q2. Post-processing for Metric Evaluation**
> We appreciate this important concern. The high PSNR values are attributed to two factors:
>
> (1) **Low acceleration factor**: The high PSNR values primarily occur at AF=3, where a large portion of k-space is retained. INR-based methods can easily achieve subject-level overfitting under such mild undersampling.
>
> (2) **Foreground ROI evaluation**: Metrics are computed on magnitude images normalized to [0, 1] within a pre-extracted ROI (brain or knee region), suppressing background voxels dominated by noise. It is a standard practice. There is no data leakage, and the train/test split is at the subject level.
>
> ## **Q3. Comparison with standard reconstruction approaches**
> We provide a comprehensive comparison of reconstruction time and computational cost for all methods in the Tbale R7.
>
> |Method|w/oIPOD|w/IPOD|GPU Memory(GB)|
> |---|---|---|---|
> |SENSE-L1W|3.5s|—|0.83|
> |SENSE-TV|15.3s|—|0.79|
> |GRAPPA|4.8s|—|0.75|
> |E2E-VN|1.3s (Infer) /16.4h (Train)|—|2.24 (Infer) / 56.4 (Train)|
> |ConvDecoder|192.5s|—|1.80|
> |SSDU|42.6s|—|8.26|
> |DINER|13.5s (Infer) / 3.1h (ML)|5.8s|1.52 (Infer) / 20.9 (ML)|
> |SIREN|116.7s (Infer) / 4.5h (ML)|37.8s|1.93 (Infer) / 24.2 (ML)|
> |HASH|11.6s (Infer) / 3.1h (ML)|4.2s|2.16 (Infer) / 21.3 (ML)|
>
> *Table R7: Average GPU reconstruction time and cost comparison across optimization-based methods, deep unfolding methods, and IPOD with different backbones.(ML: Meta-Learning)*
>
> IPOD achieves 2.3–3.1× speedup over vanilla INR baselines, making it comparable to classical methods in reconstruction time.
>
> ## **Q4. Novelty clarification**
> We respectfully clarify that our contribution extends beyond adaptive weighting:
>
> - **Not "unsupervised" rebranding**: Unsupervised methods rely on hand-crafted priors (TV, sparsity). IPOD learns a data-driven initialization via physics-based forward-model consistency — no references, no hand-crafted regularizers. This meta-objective over complex-valued multi-coil MRI data is absent in existing Reptile adaptations.
>
> - **Systematic study**: We provide the first in-depth analysis of how problem set diversity, INR backbone choice, and optimization dynamics jointly affect meta-initialization quality — design principles not available in prior work.
>
> - **Demonstrated effectiveness**: Experiments show IPOD consistently improves convergence and quality across backbones, sampling patterns, and OOD settings, validating the framework's generality.
>
> ## **Q5. Potential grammatical mistakes and typos**
> We have conducted a thorough proofreading of the entire manuscript and will incorporate all in the revised version.

---

> > ### Author Rebuttal · Reviewer_9yKf · 2026-04-01
> >
> > Thank you for this highly responsive rebuttal. The addition of the six standard clinical and deep learning baselines (GRAPPA, SENSE, E2E-VN) improves the manuscript and provides necessary context to evaluate the proposed method. I also appreciate the provided wall-clock inference times. This proves your claims of efficiency.
> > While I still question the exceptionally high PSNR scores, I thank you for the clarifications regarding the foreground ROI masking and the absence of data leakage. I still think that the theoretical machine-learning novelty is limited, the practical engineering and empirical results for medical imaging are now sound.
> >
> > I would like to remind you to ensure a dedicated 'Limitations' section is included in the camera-ready version as requested.
> >
> > Because of the rigorous rebuttal I raise my score to 4: weak accept

---

> > > ### Author Response · Authors · 2026-04-01
> > >
> > > We sincerely thank the reviewer 9yKf for the careful re-evaluation and for raising the score. We are encouraged by your recognition of our rebuttal efforts and the improved empirical soundness of the manuscript. Your constructive comments and insightful suggestions were incredibly valuable, and addressing them has significantly improved the overall quality of our paper.
> > >
> > > We will present the novelty more clearly in the revision, especially the reference-free meta-learning formulation for MRI reconstruction inverse problem and the systematic design-factor analysis. We confirm that a dedicated **Limitations** section will be included in the camera-ready version, as requested. Thank you again for your constructive feedback throughout this process.
> > >
> > > Lastly, we kindly note that the score in the system does not yet appear to reflect the updated assessment mentioned above. We would be grateful if you could update it at your convenience.

---

### Official Review · Reviewer_49o4 · 2026-03-10

**Soundness:** 2
**Presentation:** 4
**Significance:** 3
**Originality:** 3
**Overall Recommendation:** 4
**Confidence:** 4

**Summary:**

This paper proposed IPOD which is a meta-learning framework for initializing implicit neural representations (INRs) for better MRI reconstruction. The experiments demonstrate the consistent improvement of the proposed method, and the method works for both 2D and 3D MRI reconstruction.

**Compliance With Llm Reviewing Policy:**

Affirmed.

**Final Justification:**

Most of my concerns have been addressed. And I'll maintain my score at weak accept.

**Key Questions For Authors:**

**Evaluation metrics** The paper only considers the pixel-based evaluation metrics. Could the authors also consider evaluating the reconstruction quality with the perceptual metrics, like LPIPS or DISTS.

**Motivation of using undersampled data** Could the authors further clarify the rationale for using undersampled k-space data for meta-training? Because there are already lots of fully-sampled MRI dataset publicly available.

**Limitations:**

**OOD settings** Although the work considered the out-of-distribution tasks, these tasks are still limited to similar settings. It could also be interesting to consider more OOD settings, such as different anatomies, and to include pathological cases.

**Comparison with other reconstruction methods** Although the proposed method shows consistent improvements over INR-based baselines, its performance relative to other methods like supervised neural networks[1] and training-free approaches such as untrained neural networks[2] remains unclear.

[1] Sriram, Anuroop, et al. "End-to-end variational networks for accelerated MRI reconstruction." International conference on medical image computing and computer-assisted intervention. Cham: Springer International Publishing, 2020.

[2] Darestani, Mohammad Zalbagi, and Reinhard Heckel. "Accelerated MRI with un-trained neural networks." IEEE Transactions on Computational Imaging 7 (2021): 724-733.

**Strengths And Weaknesses:**

- **Soundness** This paper proposed a reference-free meta-learning framework for INR initialization, and the method is evaluated carefully and the results are also very convincing. However, the motivation for using undersampled k-space data for meta-learning is unclear, for the training process, the fully sampled data is usually available and there are lots of existing dataset and sufficient for training.
- **Presentation** This paper is good writing and well organized, making it easy to follow. And the results and tables are also clear and easy to understand.
- **Significance** The method provides a novel approach for initializing INR for better MRI reconstruction.
- **Originality** This paper provides a new insight for INR, which utilizes meta-learning for better initialization, which can not only lead to faster convergence and also better performance.

---

> ### Author Rebuttal · Authors · 2026-03-31
>
> Thank the reviewer for the valuable comments. We are encouraged by your recognition of our work. Below, we provide point-to-point responses to address your concerns.
>
> ---
> ## **Q1 & Q4. Broader baselines + perceptual metrics.**
>
> Following the reviewer's suggestions, we added E2E-VN (supervised VarNet [1]) and ConvDecoder (untrained NN [2]). Additionally, `following Reviewers Z2xB and 9yKf`, we included four more baselines:
> - **GRAPPA, SENSE-TV, SENSE-L1Wavelet** (optimization-based);
> - **SSDU** (self-supervised unfolding [Yaman et al., 2020]).
>
> |Method|AF=4|AF=8|
> |---|---|---|
> |SENSE-TV|0.0042±0.0004/0.0389±0.0008|0.1337±0.0143/0.1890±0.0138|
> |SENSE-L1W|0.0031±0.0007/0.0314±0.0039|0.1390±0.0022/0.1635±0.0059|
> |GRAPPA|0.0022±0.0007/0.0265±0.0022|0.1454±0.0153/0.1699±0.0156|
> | ConvDecoder|0.0025±0.0007/0.0276±0.0025|0.0628±0.0121/0.1385±0.0112|
> |SSDU|0.0023±0.0008/0.0268±0.0028|0.0486±0.0104/0.1259±0.0101|
> |E2E-VN(supervised)|0.0021±0.0006/0.0228±0.0020|**0.0224±0.0068**/0.1012±0.0087|
> |DINER-IPOD(Ours)|0.0022±0.0006/0.0259±0.0019|0.0323±0.0079/0.1180±0.0135|
> |SIREN-IPOD(Ours)|0.0023±0.0009/0.0261±0.0037|0.0328±0.0106/0.1201±0.0116|
> |HASH-IPOD(Ours)|**0.0020±0.0006**/**0.0211±0.0063**|0.0318±0.0081/0.1155±0.0107|
>
> *Table R5: Quantitative results (LPIPS/DISTS) of six baselines and IPOD for 1D Cartesian undersampling on fastMRI T2w and FLAIR Brain dataset.*
>
> As shown in Table R5, under AF=4 all methods achieve very low perceptual distortion (LPIPS < 0.004), indicating that mild undersampling is well-handled across paradigms. The differences become more pronounced under AF=8: classical methods show significantly higher perceptual distortion , while IPOD methods consistently outperform self-supervised and untrained baselines, approaching the supervised E2E-VN .
>
> **Full PSNR/SSIM results under AF=4 and AF=8 are presented in Table R1** (`see Q3 to Reviewer Z2xB`). IPOD methods are competitive with E2E-VN under AF=4 and outperform all non-supervised baselines under AF=8, while not requiring any fully-sampled training data. We will include these results in the revised paper.
>
> ## **Q2. Motivation of using undersampled data for meta-training.**
>
> This is a profound question. While fully-sampled datasets exist, our reference-free design is motivated by **clinical reality**:
>
> - In many scenarios — pediatric, fetal, emergency, interventional, and pregnancy MRI — scan time is strictly limited due to **patient compliance, motion, or safety constraints**, making fully-sampled acquisitions impractical.
>
> - Large-scale fully-sampled datasets are thus difficult to obtain for these populations, and training models directly on abundant fully-sampled data is often **not available in such clinical contexts**.
>
> - However, IPOD directly leverages the **abundant undersampled scans** routinely acquired in clinical practice. By meta-learning from these data, IPOD learns a **physics-consistent reconstruction prior** rather than a simple input-output mapping, making it inherently more robust across heterogeneous clinical settings.
>
> ## **Q3. More challenging OOD settings.**
>
> Following the reviewer's suggestion, we designed a strict OOD evaluation. We tested all methods on a Lesion Brain dataset with 2D Poisson undersampling (AF=10). Crucially, the Lesion data was unseen during both E2E-VN's supervised training and IPOD's meta-initialization. The Poisson sampling pattern was also absent from both training pipelines, ensuring a rigorous OOD evaluation:
>
> |Method|PSNR(↑)|SSIM(↑)|LPIPS(↓)|DISTS(↓)|
> |---|---|---|---|---|
> |SENSE-TV|30.75±1.07|0.916±0.016|0.114±0.012|0.108±0.010|
> |SENSE-L1W|30.18±1.58|0.923±0.023|0.087±0.009|0.098±0.011|
> | ConvDecoder|32.75±0.21|0.946±0.004|0.062±0.005|0.084±0.006|
> |SSDU|33.28±0.49|0.950±0.006|0.048±0.004|0.072±0.005|
> |E2E-VN(supervised)|22.95±1.00|0.788±0.020|0.248±0.010|0.216±0.020|
> |DINER-IPOD(Ours)|33.58±0.35|0.956±0.002|0.021±0.001|0.067±0.003|
> |SIREN-IPOD(Ours)|33.85±0.68|**0.958±0.002**|**0.019±0.001**|**0.064±0.003**|
> |HASH-IPOD(Ours)|**34.27±0.45**|0.957±0.002|0.020±0.001|0.066±0.002|
>
> *Table R6: Quantitative results of six baselines and IPOD for an unseen 2D Poisson sampling pattern (AF =10) on the unseen Lesion Brain dataset.*
>
> As shown in Table R6, we observe that:
> - E2E-VN suffers a catastrophic drop. Its learned priors are tightly coupled to the Cartesian trajectory seen during training; Poisson produces fundamentally different aliasing that these priors cannot handle.
>
> - All IPOD variants maintain strong performance, as test-time INR optimization adapts to observed measurements via the forward model, remaining agnostic to sampling.
>
> Qualitative comparisons in Fig. R4 (https://anonymous.4open.science/r/danjcs-2073) further confirm that IPOD preserves stable reconstruction even for small lesion regions in pathological data. This highlights IPOD's advantage: meta-learned initialization + physics-driven test-time optimization ensures robustness to distribution shifts in both anatomy and sampling.

---

> > ### Author Rebuttal · Reviewer_49o4 · 2026-04-03
> >
> > Thanks for the additional clarification and the convincing experiments, and most of my concerns have been addressed. The proposed method is competitive compared with other baselines, especially in OOD settings.

---

> > > ### Author Response · Authors · 2026-04-03
> > >
> > > We thank the reviewer 49o4 for the positive feedback and the considerable effort throughout the review process.
> > >
> > > The suggestions were highly constructive and have meaningfully improved the quality of our work.
> > >
> > > We are encouraged that most concerns have been addressed, and that the proposed method is found to be competitive, particularly in OOD settings, and we would be grateful if these improvements could be reflected in the overall evaluation.

---

### Official Review · Reviewer_xRoQ · 2026-03-10

**Soundness:** 3
**Presentation:** 3
**Significance:** 3
**Originality:** 3
**Overall Recommendation:** 2
**Confidence:** 5

**Summary:**

This paper presents a novel contribution for MRI reconstruction using a meta-learning (ML) framework, termed as IPOD. The IPOD framework allows to eliminate the dependence on the ground truth (reference) images in the meta-leaning stage. Also, an adaptive meta-update methodology is presented to guarantee optimal performance within tasks. Thus, IPOD offers an effective initialization (meta-initialization) for  the INR networks without requiring the need of ground-truth samples. The main insight for obtaining such effective initialization, is that diverse MRI inverse problems share underlying similarities in their solution manifolds. By learning from a population of inverse problems during meta-training, IPOD may capture similar patterns and embed them into initialization parameters. In this way, IPOD overcomes  two key limitations encountered in INR based methodologies, that are: 1) limited exploitation regarding  population-level data priors, and 2) poor initialization  as the INR starts from initialized networks.

**Compliance With Llm Reviewing Policy:**

Affirmed.

**Final Justification:**

I appreciate the authors’ efforts in revising the manuscript and addressing the reviewers’ comments. While the updated version includes additional experimental results, these do not fully resolve the key concerns previously raised. In particular, the paper’s core contribution would benefit from a more carefully designed and systematically analyzed experimental framework.

Specifically, the experimental setup could be strengthened by focusing on a single organ and conducting a clear comparison between single- and multi-modality scenarios, rather than mixing multiple organs. In addition, the tasks should be more clearly defined, with an explicit discussion of which tasks are meaningful and appropriate for each organ.

The evaluation would also benefit from a more comprehensive comparison against relevant baselines, including multi-armed bandits (MAB), contextual MAB (CMAB), task-adaptive schedulers, neural bandit schedulers, and teacher–student curriculum learning approaches. Furthermore, exploring learning-based task scheduling strategies could significantly enhance the contribution. For example, instead of relying on a fixed scheduler during meta-training, a neural scheduler could be introduced to dynamically predict the sampling probabilities of training tasks.

Finally, given the known sensitivity of meta-learning methods to data partitioning, the paper would benefit from a more thorough analysis of different train/validation/test splits.

Based on the above considerations, I regret that I cannot recommend acceptance of this paper in its current form.

References

Xinzhe Zuo et al., “Understanding Train–Validation Split in Meta-Learning with Neural Networks,” ICLR 2023.

Yu Bai et al., “How Important is the Train–Validation Split in Meta-Learning?” ICML 2022

**Key Questions For Authors:**

-The meta-learning framework could be better described by introducing more explanations linked with a precise analytical formulation. For instance, it is missing the objective of the meta-training process formulated as the bi-level optimization problem, and how precisely this is handled.

-The description could be enriched with more visualization figures, for instance, the shared underlying similarities in the manifolds. How this impacts the ML framework is also interesting to be investigated. More specifically, when samples come from the different modalities/organs/devices, how does this impact the manifold similarity …

-I can understand the inner loop as described in Sec. 3.2. However, the outer loop is not addressed, it appears to be missing. This enforces what I already described earlier: analytical and precise details regarding  inner and outer loop are mandatory.

-Another issue is related to tasks used in the inner loop. It  is known that task clashing can occur in the meta-learning process. This is somehow related when the authors state in the paper (Sec. 3.2) “Specifically, when confronted with challenging reconstructions, the optimization may converge to poor local minima or exhibit unstable convergence behavior, resulting in network parameters that inadequately represent the underlying image structure”. This is precisely due to the nature of the tasks that play different importance in the meta-training. As such gradient interference also may occur within tasks, jeopardizing the overall meta-training stage.

-Also, in this paper, the authors randomly sample the tasks. Having the assumption behind such uniform sampling, requires that  the tasks are equally important -  which is often not the case. First, some tasks could be noisy, second, the number of meta-training tasks is likely limited, so that the distribution over different clusters of tasks is uneven.

-To address the issues above, a task scheduling strategy is crucial to be incorporated in the ML framework, which is missing. Specifically, equip the meta-learning framework with a task scheduler that determines which tasks should be used for meta-training in the current iteration. Although the authors assign different weights to different tasks, this is performed under the umbrella of uniform task scheduling.  So, one may wonder, is it really necessary to adapt weights when we have a good task scheduler ? ..

-Also, related with adaptive weighting, it is not convincing, having the eq. (7) to handle such complex problems. Instead of assigning the weights to the tasks, having, for instance,  a neural scheduler to predict the probability of each training task being sampled would be welcome. With such a strategy we could have a task scheduler into meta-learning that would adapt to the progress of the meta-model.

-The experimental setup constitutes a big jump.

It would be more beneficial to have progress in the experiments and to ascertain the performance of the ML framework. It is known that the ML has different convergence behavior depending on the tasks (and also the task scheduler). Having a setup, containing only one organ and having different modalities (intra-task organ) could be a 1st step. Then, incorporating several organs (inter-task organs), could be a more ambitious step. For instance, it would be welcome a more organized section of expetrimantelç results, namely:

Define the tasks and its corresponding scheduling for problems with one organ with one single modality.

Define the tasks and its corresponding scheduling, but now with several modalities. Also, we must define the tasks and schedule.

More challenging is to have several organs with several modalities.

As it is, it is very difficult to judge the results in Table 1.

And of course, from the above, several issues arise, namely: What are precisely the tasks ? What are the dominant and weaker tasks ? What is the impact of the train-validation split ? (as this is a crucial issue in ML to learn a good prior model)

**Limitations:**

Please see the "Key Questions For Author" section.

**Strengths And Weaknesses:**

Strengths :

-The application of a meta-learning (ML) framework to MRI reconstruction is an interesting and promising aspect of this work. Leveraging meta-learning for inverse problems in medical imaging is a relevant research direction and fits well within the scope of the paper.

-The proposed IPOD framework provides a perspective on inverse-problem-driven learning. In particular, the possibility of obtaining a robust initialization for implicit neural representations (INRs) without requiring fully sampled ground-truth data is an appealing property of the method.

Weaknesses:

-The meta-learning component could be described in greater depth. In particular, the analytical formulation and the optimization procedure could benefit from additional clarification to help readers better understand the role of meta-learning in the proposed framework.

-A more detailed discussion of the task scheduling strategy would strengthen the paper. Providing additional motivation, design choices, or ablation studies regarding how tasks are scheduled could help clarify its contribution to the overall performance.

-It is somewhat difficult to clearly identify where the main performance gains originate. The experimental setup could be further expanded to analyze the role of task complexity and its interaction with the proposed approach, which would provide better insight into the advantages of the method.

---

> ### Author Rebuttal · Authors · 2026-03-31
>
> Thanks for your efforts and valuable comments. Below, we provide point-to-point responses to address your concerns.
> ---
> ## **Q1. Bi-level optimization formulation and outer loop.**
>
> Our framework naturally follows a bi-level structure:
> - the inner loop treats individual MRI sample as a separate inverse problem;
>
> - the outer loop aggregates parameter updates across samples to obtain the meta-initialization.
>
> - the outer-loop formulation is presented in the "Adaptive Weighted Meta-optimization" section of our paper.
>
> We will provide a more explicit description of the bi-level in the revision.
>
> ## **Q2. Diversity of Problem Set $\boldsymbol{S}$ and its impact on generalization.**
> **Our problem set $\boldsymbol{S}$ already spans multiple organs and modalities.** (`see Q1 of Reviewer Z2xB`). Furthermore, we designed a progressive ablation to isolate the effect of each diversity axis:
> - **IPOD-Cartesian—multiple anatomies, single sampling**: Meta-training with Cartesian-only undersampling.
> - **IPOD-Brain—single anatomy, multiple samplings**: Meta-training on Brain-only.
> - **IPOD-All—multiple anatomies, multiple samplings**: Meta-training on all dataset.
>
> Tables R2&R3 demonstrate a clear trend:
> - Increasing problem-level diversity consistently improves generalization.
>
> - On cross-forward-model transfer, IPOD-All outperforms IPOD-Cartesian by 1.3 dB.
>
> - On cross-anatomy transfer, IPOD-All outperforms IPOD-Brain by 1.6 dB.
>
> We also visualize the reconstruction process at progressive iterations for these ablation settings in **Fig. R1&R2 (https://anonymous.4open.science/r/danjcs-2073)**, which further illustrate the effect of the diverse problem set.
>
> |Method|Iter10||Iter100||Iter500||Iter1000||
> |---|---|---|---|---|---|---|---|---|
> ||PSNR|SSIM|PSNR|SSIM|PSNR|SSIM|PSNR|SSIM|
> |w/oIPOD|12.74±0.98|0.510±0.056|23.90±1.87|0.772±0.006|29.17±0.31|0.891±0.017|31.47±0.45|0.934±0.006|
> |IPOD-Cartesian|16.40±4.36|0.530±0.017|26.80±0.84|0.832±0.011|31.91±0.25|0.927±0.002|32.92±0.79|0.953±0.008|
> |IPOD-All|18.51±1.00|0.604±0.022|30.97±0.99|0.907±0.009|33.78±1.76|0.957±0.007|34.21±2.96|0.968±0.006|
>
> *Table R2: Quantitative comparison of Baselines and IPOD with different meta-learning strategies for radial undersampling.*
>
> |Method|Iter10||Iter100||Iter500||Iter1000||
> |---|---|---|---|---|---|---|---|---|
> ||PSNR|SSIM|PSNR|SSIM|PSNR|SSIM|PSNR|SSIM|
> |w/oIPOD|13.81±2.62|0.539±0.057|26.59±3.11|0.833±0.050|29.99±2.81|0.904±0.026|31.71±2.83|0.929±0.018|
> |IPOD-Brain|17.74±1.08|0.667±0.042|29.33±3.53|0.899±0.040|31.91±3.68|0.937±0.028|33.14±3.38|0.950±0.023|
> |IPOD-All|19.57±2.58|0.726±0.044|30.72±3.35|0.919±0.029|33.04±3.93|0.951±0.020|34.74±3.58|0.960±0.017|
>
> *Table R3: Quantitative comparison of Baselines and IPOD with different meta-learning strategies on fastMRI Knee Dataset.*
>
> ## **Q3. Task scheduling and adaptive weighting.**
>
> We address the concerns about task scheduling and adaptive weighting from the following perspectives:
>
> - **Balanced problem set**: $\boldsymbol{S}$ covers clinically common contrasts and anatomies — none of which should be deliberately categorized as primary or secondary tasks.
>
> - **Good prior in model**: Low-AF tasks provide stronger priors , but deliberately prioritizing them would bias the initialization toward easy regimes and hurt generalization to unseen scenarios (e.g., high-AF, unseen sampling patterns).
>
> - **Adaptive weighting**: Our weighting is a simple yet effective mechanism that down-weights poorly converging tasks to mitigate gradient interference, as confirmed by the ablation study in our paper.
>
> - **Neural task scheduler**: We appreciate this suggestion. A learned scheduler that dynamically adjusts task sampling probabilities is a promising direction for future research and will be discussed in the revision.
>
> ## **Q4. Training–Validation Split**
>
> - **Subject-level split**: No slices from the same patient appear in both meta-training and evaluation sets.
>
> - **Uniform task distribution**: Each task in $\boldsymbol{S}$ contains the same number of samples.
> ## **Q5. Optimization trajectory distance.**
> To quantify the effect of meta-initialization, we measure ‖θ_t − θ_0‖₂ throughout training over test samples, where θ_i denotes the model parameters at the i-th epoch.
>
> - **Smaller total drift**: Meta-Init drifts ~1.49× less than Random-Init , indicating the meta-learned starting point is already closer to the solution manifold.
>
> - **Faster stabilization**: The per-epoch increment (Δmean) of Meta-Init decreases rapidly, suggesting quick convergence near the target.
>
> |Epoch|Random-Init|Δmean(Random)|Meta-Init|Δmean(Meta)|
> |---|---|---|---|---|
> |0|0.0000±0.0000|—|0.0000±0.0000|—|
> |100|0.6248±0.0041|0.6248|0.5633±0.0340|0.5633|
> |200|0.7598±0.0173|0.1350|0.6361±0.0464|0.0728|
> |400|0.9203±0.0418|0.1605|0.7002±0.0567|0.0641|
> |800|1.0910±0.0788|0.1707|0.7550±0.0616|0.0548|
> |1000|1.1422±0.0933|0.0512|0.7689±0.0615|0.0139|
>
> *Table R4: L2 parameter drift during optimization for Random-Init vs. Meta-Init.*

---

> > ### Author Rebuttal · Reviewer_xRoQ · 2026-04-01
> >
> > I thank the authors for providing the feedback for the questions raised. However, I’m not (sufficiently) convinced. I still think that the theoretical machine-learning novelty is somewhat limited, and to enrich the theoretical content some issues are mandatory to include. I would expect to see all the (detailed) theoretical details in this rebuttal. Also, a nuclear aspect of meta-learning is related to task scheduling (TS), as mentioned in my previous review. Concretely, the paper treats each task as equally important though randomly sampling strategy. However, this assumption can possibly fail in real-world scenarios. That is why other important schedules should be investigated. The manuscript lacks a study regarding this use since there is a large body of work concerning TS that is proposed to  “refine” the meta learning strategy, i.e. (1) methods working on the gradient that updates trainable parameter  have proven their superiority over meta-training methods with uniform sampling, (2) Curriculum learning, (3) methods based on task adaptation difficulty, (4) Contextual Multi-Arm Bandit, (5) neural bandit scheduler. Although the proposed methodology is simple, my feeling is the results can be investigated against a TS methodology. Also note that, the "controlled" experimental setup scenarios (for progressive ablation), is necessary. Regarding this issue I would expect to see results taking in consideration some of my concerns, and for each case what is the importance within tasks for each experimental scenario.  As it is, it is  difficult to ascertain  the level of difficulty of the tasks, as well as  the gradient clash that may occur.
> > Another aspect that is not sufficiently detailed is the training/validation/test split. Please note that this is another issue that impacts the performance of the meta-learning

---

> > > ### Author Response · Authors · 2026-04-06
> > >
> > > Thank you for your follow-up and patience. We have conducted experiments to address your concerns.
> > > ---
> > > ## **Q1.Clarification of Dataset Split**
> > > In the manuscript, we provided a detailed description of train and evaluation protocols. For better clarity, we summarize them below.
> > >
> > > |Type|Setting|Dataset|Contrast|Anatomy|Undersampling|
> > > |---|---|---|---|---|---|
> > > |Meta-Training|-|fastMRI|T2w,FLAIR|Brain,Knee|Cartesian,Random|
> > > |Valid|Cross-contrast|fastMRI|T1w,T2w,FLAIR|Brain,Knee|Cartesian|
> > > |Valid|Cross-dataset|MoDL|T2w|Brain|Cartesian|
> > > |Valid|Cross-physical model,dataset|fastMRI,MoDL|T2w|Brain|Radial|
> > > |Valid|Cross-undersampling,dataset|fastMRI,MoDL|T1w,T2w,FLAIR|Brain,Knee|Poisson,Gaussian|
> > > |Valid|Cross-contrast,dataset|In-house(Prospective)|T1w|Brain|Cartesian|
> > > |Valid|Cross-contrast,undersampling,dataset|In-house(3D)|T1w|Brain|Cartesian,Poisson,Gaussian|
> > >
> > > *Table S1.Data splits.*
> > >
> > > ## **Q2.Task Scheduling(TS)**
> > > Experiments with identical settings except task sampling:
> > > - **IPOD**: Uniform sampling + adaptive weighting.
> > > - **Predefined Curriculum (CL)**: Easy → Easy+Medium → All.
> > > - **Loss-Proportional (LP)**: Sampling Probability ∝ loss.
> > > - **Inverse-Loss (IL)**: Sampling Probability ∝ 1/loss.
> > >
> > > Difficulty levels: Easy(AF=2/5 for Cartesian/Random undersampling), Medium(AF=4/10), Hard(AF=6/15).
> > >
> > > |Method|PSNR|SSIM|
> > > |---|---|---|
> > > |IL|41.26±0.96|0.994±0.005|
> > > |LP|**42.07±0.47**|**0.998±0.004**|
> > > |CL|*41.93±0.65*|*0.998±0.006*|
> > > |IPOD|41.82±0.72|0.998±0.007|
> > >
> > > *Table S2.fastMRI(AF=4 Cartesian)*
> > >
> > > |Method|PSNR|SSIM|
> > > |---|---|---|
> > > |IL|32.98±2.02|0.932±0.060|
> > > |LP|32.91±2.01|0.925±0.059|
> > > |CL|*33.77±2.29*|*0.935±0.019*|
> > > |IPOD|**34.55±2.86**|**0.947±0.005**|
> > >
> > > *Table S3.fastMRI(AF=8 Cartesian)*
> > >
> > > |Method|PSNR|SSIM|
> > > |---|---|---|
> > > |IL|32.42±0.31|0.937±0.004|
> > > |LP|33.23±0.42|0.942±0.002|
> > > |CL|*33.86±0.29*|*0.955±0.001*|
> > > |IPOD|**34.27±0.45**|**0.957±0.002**|
> > >
> > > *Table S4.Lesion(AF=10 Poisson)*
> > >
> > > |Method|PSNR|SSIM|
> > > |---|---|---|
> > > |IL|33.64±1.22|0.954±0.006|
> > > |LP|34.25±1.43|0.959±0.005|
> > > |CL|*34.53±1.29*|*0.959±0.005*|
> > > |IPOD|**35.18±0.84**|**0.963±0.008**|
> > >
> > > *Table S5.MoDL(AF=12 Radial)*
> > >
> > > We provide key points for the above:
> > > - **K-space Domain.** Our loss enforces physical consistency in k-space via the forward model. Task difficulty derived from k-space does not necessarily align with difficulty in the image domain, which may limit the effectiveness of heuristic sampling strategies.
> > > - **Bias from LP/IL.** Because LP/IL derive sampling probabilities directly from loss, they may over-represent specific subsets of tasks, leading to an imbalanced coverage and suboptimal meta-initialization.
> > > - **CL Analysis.** CL stabilizes training via an easy-to-hard schedule. However, in k-space reconstruction inverse tasks, this staged progression may bias optimization toward dominant regions of the task manifold, potentially resulting in insufficient coverage, and consequently weaker generalization under complex settings.
> > >
> > > ## **Q3.Task Interference**
> > > We compute $\Delta_i=\theta_{{adapted},i}-\theta_{{meta}}$ after inner-loop adaptation and measure pairwise cosine similarity. Clash ratio=fraction of negative pairs.
> > >
> > > |Method|Ep100||Ep500||Ep1500||Ep2000||
> > > |---|---|---|---|---|---|---|---|---|
> > > ||CoS|Clash|CoS|Clash|CoS|Clash|CoS|Clash|
> > > |IL|0.721±0.067|0.0|0.699±0.074|0.1|0.633±0.073|0.4|0.610±0.071|0.9|
> > > |LP|0.739±0.061|0.0|0.705±0.070|0.1|0.659±0.074|0.3|0.613±0.073|0.7|
> > > |CL|0.734±0.059|0.0|0.683±0.070|0.1|0.636±0.072|0.4|0.591±0.075|1.5|
> > > |IPOD|0.763±0.069|0.0|0.553±0.103|1.8|0.347±0.133|2.3|0.269±0.172|6.6|
> > >
> > > *Table S6. Cosine similarity(CoS) and clash(%) of task gradients.*
> > >
> > > Two key points:
> > > - **Initial alignment:** All methods start with high cosine and negligible clash, reflecting shared reconstruction priors.
> > > - **Late-stage divergence:** IPOD shows lower similarity and higher clash, indicating refined capacity for task-level decoupling in k-space domain.
> > >
> > > All discussion on TS will be included in the revision.
> > >
> > > ## **Q4. Theoretical Details**
> > > **Due to space limits, we highlight the core mechanism and welcome further discussion.**
> > >
> > > IPOD is a bi-level optimization on the parameter manifold:
> > > - **Inner loop.** Each task $\boldsymbol{S}_n$ initializes $\boldsymbol{\Theta}_n\leftarrow\boldsymbol{\Theta}_m$ and solves its inverse problem: $\boldsymbol{\Theta}_n\leftarrow\boldsymbol{\Theta}_n-\gamma\nabla\mathcal{L}_n$, with $\mathcal{L}_n=\frac{1}{2}\|\boldsymbol{S}_n-\mathcal{A}_n\boldsymbol{I}_n\|_2^2+\lambda\|\mathbf{G}\boldsymbol{I}_n\|_1$. The displacement $\boldsymbol{\Theta}_n-\boldsymbol{\Theta}_m$ is the gradient encoding the task's optimization trajectory.
> > > - **Outer loop.** Trajectories are aggregated via adaptive weighted Reptile: $\boldsymbol{\Theta}_m\leftarrow\boldsymbol{\Theta}_m+N\alpha\sum w_n(\boldsymbol{\Theta}_n-\boldsymbol{\Theta}_m)$, where $w_n\propto 1/\mathcal{L}_n$, steering the meta-prior toward the geometric centroid of the task solution manifold.
> > >
> > > ## **Q5.Progressive ablation**
> > > Results(anatomy and undersampling) can be found in the original Q2.

---

### Official Review · Reviewer_Z2xB · 2026-03-13

**Soundness:** 3
**Presentation:** 4
**Significance:** 3
**Originality:** 2
**Overall Recommendation:** 4
**Confidence:** 4

**Summary:**

This paper focuses on MRI reconstruction with implicit neural representations (INR). It proposes a reference-free meta-learning framework, IPOD, to learn initialization parameters directly from undersampled measurements. The method is designed to improve the adaptation speed and reconstruction quality of INR-based models on new MRI inverse problems. Experiments are conducted on multiple datasets and sampling settings. This paper reports improved convergence speed and reconstruction performance over several INR baselines.

**Compliance With Llm Reviewing Policy:**

Affirmed.

**Final Justification:**

I believe my comments have been addressed. That said, now that I have a fuller picture of the paper and its contributions, I find the contribution and novelty sufficient, though not exceptional. I will keep my score unchanged, as I view this paper as solid but relatively routine.

**Key Questions For Authors:**

- Does the problem set $S$ involve different dataset and different sampling pattern?

Please also refer to the weakness section.

**Limitations:**

Limitation is not discussed in this paper.

**Strengths And Weaknesses:**

Pro:
- The paper explores an interesting and relatively less studied direction. This provides a potentially useful perspective for MRI reconstruction.
- The paper is generally well presented. The figures are clear and help illustrate both the overall framework and the experimental results.
- The reported empirical results are strong.

Con:
- The main technical contributions are not sufficiently clear. From the current presentation, several core components appear to be based on existing ideas, including meta-learning, a weighted variant of Reptile-style updates, and INR-based reconstruction. As a result, it is difficult to identify what the central methodological novelty is and which part should be regarded as the primary technical contribution of the paper.
- For Cartesian sampling, the reported AF=3/4 appears relatively small. I wonder how the proposed method performs under more challenging Cartesian undersampling regimes.
- I understand that this paper is in the context of INR, but I think it is also important to showcases baseline results that are based on non-INR but widely-used methods, such as diffusion model-based, deep unfolding, optimization-based algorithms. It would help reader to better position this paper among the large volume of MRI reconstruction literatures.

---

> ### Author Rebuttal · Authors · 2026-03-30
>
> We thank the reviewer for the valuable comments. Below, we provide detailed point-by-point responses.
>
> ---
> ## **Q1.  Does the problem set $\boldsymbol{S}$ involve different dataset and different sampling pattern?**
> Our meta-training is conducted solely on the fastMRI dataset, without incorporating additional external datasets. However, the problem set $\boldsymbol{S}$ is constructed to be diverse along four axes:
> - **multiple anatomies** (Brain, Knee);
> - **multiple sampling patterns** (1D Cartesian, 2D Random);
> - **multiple contrasts** (T2w, FLAIR);
> - **multiple acceleration factors**.
>
> Each unique combination of the above defines a distinct inverse problem (task). We have also conducted ablation experiments to verify the impact of problem-level diversity within $\boldsymbol{S}$ on the generalization of the meta-initialization (`see Q2 of Reviewer xRoQ`).
>
> ## **Q2. Novelty clarification.**
> We appreciate this feedback. Rather than a simple combination of existing components, our core contribution is proposing a general and practical framework for learning transferable INR initializations from undersampled MRI data alone, and conducting an in-depth study of the key factors that govern its effectiveness:
>
> (1) **Problem formulation**: We formulate INR initialization as a meta-learning problem over a population of MRI inverse problems. Unlike prior meta-INR work which focuses on visual tasks with supervised losses, MRI reconstruction involves complex-valued computations, diverse anatomies, and multi-center variability — making the extension non-trivial.
>
> (2) **Reference-free meta-objective**: Our outer-loop is driven entirely by physics-based data-consistency losses — no fully-sampled references needed. This reformulates the meta-objective around the MRI forward model, a departure from conventional meta-learning.
>
> (3) **Architecture generalization + diversity-robustness**: We systematically evaluate three INR backbones, establishing architecture-specific trade-offs. Our ablation studies further reveal that diverse problem sets create more robust priors — a design principle not studied in prior work.
>
> We will make this contribution structure more explicit in the revision.
>
> ## **Q3. Harder Cartesian AF + broader non-INR baselines.**
> We agree with both suggestions. We extended the Cartesian evaluation to AF=8 and added six representative non-INR methods spanning different paradigms:
> - **GRAPPA, SENSE-TV, SENSE-L1Wavelet** (classical optimization-based);
> - **ConvDecoder** (untrained NN [Darestani & Heckel, 2021], `suggested by Reviewer 49o4`);
> - **E2E-VN** (supervised VarNet [Sriram et al., 2020], `also suggested by Reviewer 49o4`);
> - **SSDU** (self-supervised unfolding [Yaman et al., 2020]).
>
> As shown in Table R3, we observe that:
> - Under both AF=4 and AF=8, all three IPOD variants outperform self-supervised and untrained baselines as well as classical methods.
>
> - At AF=4, where more k-space data is acquired, subject-level overfitting allows INR to capture fine details — DINER-IPOD performs on par with the fully supervised E2E-VN.
>
> - At AF=8, IPOD methods still approach E2E-VN while maintaining clear advantages over all other baselines.
>
> Qualitative comparisons of IPOD reconstructions against all six baselines are provided in **Fig. R3 (https://anonymous.4open.science/r/danjcs-2073)**. We also report **additional perceptual metrics (LPIPS&DISTS)** under the more difficult AF=8 setting in Table R5 (`see Q1 to Reviewer 49o4`).
>
> In particular, we evaluate all methods on a challenging OOD task — an unseen Lesion Brain dataset with 2D Poisson sampling (AF=10), where the supervised E2E-VN exhibits significant degradation while IPOD remains robust (`see Q3 of Reviewer 49o4 `). We will include these studies in the revised paper.
>
> |Method|AF=4|AF=8|
> |---|---|---|
> |SENSE-TV|36.10±2.45/0.983±0.016|25.32±4.35/0.927±0.013|
> |SENSE-L1Wavelet|37.08±0.81/0.988±0.008|26.96±1.24/0.928±0.006|
> |GRAPPA|36.31±0.76/0.989±0.007|25.07±4.52/0.926±0.014|
> |ConvDecoder|38.61±0.70/0.991±0.006|32.43±2.05/0.932±0.013|
> |SSDU|39.66±0.63/0.993±0.005|34.27±1.78/0.938±0.011|
> |E2E-VN(supervised)|41.95±0.04/**0.998±0.004**|**37.36±1.46**/0.939±0.007|
> |DINER-IPOD(Ours)|**42.01±1.28**/0.997±0.002|34.23±2.73/0.943±0.005|
> |SIREN-IPOD(Ours)|41.23±0.43/**0.998±0.009**|35.47±1.51/0.940±0.001|
> |HASH-IPOD(Ours)|41.82±0.72/**0.998±0.007**|34.55±2.86/**0.947±0.005**|
>
> *Table R1: Quantitative comparison (PSNR/SSIM) of optimization-based methods, deep unfolding methods, and IPOD with different network backbones for 1D Cartesian undersampling on fastMRI T2w and FLAIR Brain dataset.*
>
> ## **Q4. Further Discussion on Limitations**
> We will add a limitations section in the revision, including but not limited to: (1) incorporating textual priors (e.g., imaging protocols) for multi-modality reconstruction; (2) extending to more anatomical structures; (3) neural scheduler architectures for adaptive meta-training.

---

> > ### Author Rebuttal · Reviewer_Z2xB · 2026-04-04
> >
> > Thanks for the authors’ rebuttal. I believe my comments have been addressed. That said, now that I have a fuller picture of the paper and its contributions, I find the contribution and novelty sufficient, though not exceptional. I will keep my score unchanged, as I view this paper as solid but relatively routine.

---

> > > ### Author Response · Authors · 2026-04-06
> > >
> > > We thank the reviewer for the constructive and insightful comments, as well as the considerable effort throughout the review process. We are glad to hear that our response has adequately addressed all your concerns. It is encouraging to receive recognition of our contribution and novelty. Your suggestions have been highly valuable and have meaningfully improved the quality of our work.

---

### Decision · Program_Chairs · 2026-04-30

**Decision:**

Accept (regular)

**Comment:**

This submission addresses efficient MRI reconstruction by proposing a reference‑free meta‑learning framework for learning initialized and generalized parameters for implicit neural representations (INR) from under‑sampled data, with experiments demonstrating fast adaptation and high‑fidelity reconstruction across diverse imaging protocols. Reviewers mainly raised concerns about the clarity of technical contributions, comparisons with non‑INR baselines, details on the meta‑learning framework, need for more visualization figures, uneven task distribution and scheduling strategy, experimental setup, requirements for additional out‑of‑distribution evaluations, and discussions on limitations. Three of four reviewers were satisfied with the authors’ responses and maintained or increased their scores to weak accept. Reviewer xRoQ acknowledged the revisions but raised remaining concerns, particularly about the need for stronger experimental setup focusing on a single organ, comparisons between single‑ and multi‑modality scenarios. These remaining suggestions help improve the manuscript quality and should be incorporated in the revisions, but may not be the key reasons for rejection. Considering the overall recommendations towards weak accept, the paper is recommended for weak acceptance, with the suggestions that the authors should include revisions based on all the reviewers' comments and remaining concerns in the final version.